# M²F-PINN: A Multi-Scale Frequency-Domain Multi-Physics-Informed Neural Network for Ocean Forecasting

## Abstract

Physics-informed neural networks (PINNs) embed physical laws into data-driven learning and are becoming increasingly influential in climate and ocean forecasting. Yet effectively capturing multi-scale variability across high and low frequencies while maintaining training stablility and ensuring convergence remains challenging for conventional PINNs. We introduce M²F-PINN, a novel Transformer-based multi-scale frequency-domain multi-PINN algorithm designed to 1) mitigate spectral bias via Fourier representation learning, and 2) analyze multi-scale characteristics through frequency-domain modeling, and 3) incorporate physics priors using multiple PINNs. M²F-PINN leverages multi-scale Fourier networks to learn spectral components and multi-scale interactions, and employs a 3D Swin Transformer in an autoregressive setting to capture spatiotemporal regularities. The advantages of M²F-PINN include: 1) adaptively learns multi-scale frequency components to enhance the modeling of multi-scale dynamics; 2) jointly estimates physical coefficients within the PINN modules, refining representations of physical processes; 3) preserves the Transformer framework, enabling compatibility with diverse architectures and structural decoupling; 4) extensive experiments on real-world ocean datasets show that M²F-PINN outperforms deep-learning baselines and competitive ocean models (e.g., XiHe, WenHai) in predicting ocean current fields, achieving superior performance across multiple time horizons.

## 1 Introduction

The ocean serves as both a reservoir and regulator of energy within the Earth's climate system. Oceanic currents constitute the primary form of seawater movement, driven by multiple factors including wind forces, the Coriolis effect generated by Earth's rotation, variations in seawater density, and the distribution of landmasses and oceans. However, the prediction of oceanic currents across global regions remains insufficiently accurate and inefficient, thereby impacting climate forecasting.

While traditional numerical methods possess distinct advantages in terms of physical consistency and interpretability, they have limitations in predicting ocean flow fields. For instance, due to simplified parameterizations and computational constraints, traditional numerical models face bottlenecks in coupling multi-scale ocean processes and exhibit low computational efficiency. Deep learning-based methods have emerged prominently and are widely applied in meteorological and ocean forecasting, owing to their flexibility, strong adaptability, and high computational efficiency. In the field of meteorological forecasting, models such as FourCastNet Pathak et al. (2022), GraphCast Lam et al. (2023), and Pangu Weather Bi et al. (2023) have all demonstrated excellent predictive performance. On the other hand, in the more complex ocean domain, ocean forecasting models including AI-GOMS Xiong et al. (2023), XiHe Wang et al. (2024), and WenHai Cui et al. (2025) have exhibited powerful end-to-end prediction capabilities, yet they still lack consideration of ocean physical dynamic processes and suffer from insufficient interpretability. Subsequently, there have also been studies focusing on physics-guided loss functions (e.g., the LangYa model Yang et al. (2024)), accurate prediction of storm surges Zhu et al. (2025), and accurate prediction of tsunami wave fields Someya & Furumura (2025). However, the aforementioned studies still suffer from three key limitations: insufficient prediction accuracy, inadequate performance in capturing ocean multi-scale dynamic fields, and insufficient utilization of ocean frequency-domain information.

Ocean current field data contain not only long-time-series temporal information but also abundant frequency-domain information. Therefore, frequency-domain information learning is also a necessary approach to improve prediction accuracy. There have been numerous studies on the application of deep learning in the frequency domain and for high-frequency data. Tancik et al. Tancik et al. (2020) proposed a Fourier feature network, which transforms input data into a combination of sine and cosine periodic functions. This enables the neural network to learn high- and low-frequency information separately, thereby effectively addressing the spectral bias issue when learning high-frequency information. For low-resolution observational data, Shaopeng Li et al. Li et al. (2025) developed a Frequency-Domain Physics-Informed Neural Network (PINN) to more accurately predict the 3D spatiotemporal wind fields of wind turbines. Chao Song et al. Song & Wang (2023) used PINN with embedded Fourier features to simulate multi-frequency seismic wavefields. Recent work on Neural Tangent Kernel (NTK) theory has shown that the components corresponding to larger eigenvalues in the objective function of deep learning neural networks generally exhibit higher convergence rates, while eigenvalues decrease rapidly as the frequency of the objective function increases Jacot et al. (2018); Rahaman et al. (2019); Zhi-Qin et al. (2020). This reveals that neural networks always tend to learn low-frequency patterns first, followed by the remaining components.

To address the aforementioned issues, we proposes M$^2$F-PINN, a multi-scale frequency-domain multi-PINN method, for the accurate prediction of ocean current fields. M$^2$F-PINN learns the spectral features of multi-scale data via Fourier representation learning, and incorporates physical priors using PINN. This not only ensures physical consistency but also endows the method with strong physical interpretability. We verify M$^2$F-PINN's superior performance through experiments with different prediction horizons. **Our contribution** We summarize the contribution of this work. **Framework** M$^2$F-PINN integrates a Transformer backbone with a masking strategy to enable long-horizon prediction of oceanic variables. By jointly modeling temporal dynamics and spectral structure, the framework delivers accurate forecasts across timescales. **Constraints** PINN constraints are incorporated with uncertainty-aware adaptive weighting of multiple PDE-based losses, enhancing physical consistency and interpretability while capturing abrupt, event-like variations in ocean time series. **Multi-frequency** A multi-scale frequency-domain module with a tunable Fourier mapping learns projection matrices and scale parameters end-to-end, mitigating spectral bias and improving the representation of both low- and high-frequency components in ocean currents.

## 2 PRELIMINARIES

**Fourier feature embeddings.** PINNs are known to learn low frequencies first (spectral bias). We enrich the coordinate encoding with Gaussian Fourier features to expose higher frequencies to the network: Sample rows $b_\ell^\top \in \mathbb{R}^{1 \times d}$ i.i.d. from $\mathcal{N}(0, \sigma^2 I_d)$ and define, for $x \in \mathbb{R}^d$,

$$\gamma_\sigma(x) = \frac{1}{\sqrt{m}} \left[ \cos(Bx) \,\|\, \sin(Bx) \right] \in \mathbb{R}^{2m}, \quad B = \begin{bmatrix} b_1 & b_2 & \cdots & b_m \end{bmatrix}. \quad (1)$$

In expectation this induces an Radial Basis Function(RBF) kernel,

$$\mathbb{E}[\langle \gamma_\sigma(x), \gamma_\sigma(x') \rangle] = \exp\left( -\frac{1}{2} \sigma^2 \|x - x'\|_2^2 \right) \quad (2)$$

with bandwidth $\ell = 1/\sigma$. Larger $\sigma$ sharpens locality and increases the representation of high-frequency content. We use multi-scale embeddings by concatenating $\{\gamma_{\sigma_\ell}\}_{\ell=1}^L$ (small $\sigma_\ell$ for global/low-frequency, large $\sigma_\ell$ for local/high-frequency), optionally learning per-scale amplitudes and the projection matrix $B$ to adapt to oceanic spectra.

**NTK view of spectral bias.** Let $f_\theta(X)$ denote network outputs at the $N$ training inputs $X = \{x_i\}_{i=1}^N$. Under the standard infinite-width approximation, the training dynamics linearize:

$$\frac{d}{dt} f_\theta(X) \approx -K(X, X) \left( f_\theta(X) - Y \right), \quad (3)$$

with the NTK Gram matrix $K_{ij} = \left\langle \frac{\partial f_\theta(x_i)}{\partial \theta}, \frac{\partial f_\theta(x_j)}{\partial \theta} \right\rangle$, and Y is the training objective. Diagonalizing $K = Q^\top \Lambda Q$ shows that the error along eigenvector $q_k$ decays as $\exp(-\lambda_k t)$; small $\lambda_k$ components converge slowly, which constitutes spectral bias when high-frequency eigenvectors align with small eigenvalues. This lens will be used to interpret the effect of Fourier features and our multi-scale design on convergence across spatial frequencies.

# 3 PROPOSED M²F-PINN

## 3.1 GENERAL FRAMEWORK OF M²F-PINN

Here, we develop a bivariate autoregressive neural network built on the Swin Transformer architecture. The inputs are the oceanic eastward current velocity ($U$) and ocean northward current velocity ($V$). The model first performs downsampling within an encoder to extract hierarchical features, followed by upsampling in a decoder to reconstruct the outputs, thereby enabling representation learning from the data, as illustrated in Figure 1. The training algorithm for ocean forecasting with M²F-PINN is implemented as shown in Algorithm 1.

---

**Algorithm 1** Training algorithm for ocean forecasting with M²F-PINN

---

1: **Input:** Preprocessed ocean dataset $\mathcal{D}$, initial model $f_\theta$, and hyperparameters
2: **Output:** Optimized model parameters $\theta$
3: **hyperparameters:** Training epochs $E$, learning rate $lr$, weight for each variables $[U, V]$, Fourier feature parameters $FF_{\text{low}}$, $FF_{\text{high}}$, and processor $FF_{\text{processed}}(hidden\_dim)$
4: Load ocean dataset and construct dataloader $\mathcal{B}$
5: Initialize model $f_\theta$, optimizer $\mathcal{O}$, scheduler $\mathcal{S}$, parameters of Fourier features, precompute coordinate grid and static Fourier features
6: **for** epoch $= 1$ to $E$ **do**
7:     **for** each batch $(x, y_{\text{true}})$ in $\mathcal{B}$ **do**
8:         $x_{\text{ff\_low}} \leftarrow FF_{low}(\text{coordinates})$
9:         $x_{\text{ff\_high}} \leftarrow FF_{high}(\text{coordinates})$
10:         $x_{\text{ff\_processed}} \leftarrow FF_{processor}([x_{\text{ff\_low}}, x_{\text{ff\_high}}])$
11:         $x_{\text{enhanced}} \leftarrow \text{concat}(x, x_{\text{ff\_processed}})$
12:         $y_{\text{pred}} \leftarrow f_\theta(x_{\text{enhanced}})$
13:         $\mathcal{L}_{\text{data}} \leftarrow \text{WeightedMSE}(y_{\text{pred}}, y_{\text{true}})$           ▷ According to Equation 33
14:         $\mathcal{L}_{\text{PDE-UV}} \leftarrow \text{MSE}(\text{physical\_informed}(y_{\text{pred}}))$    ▷ According to Equations 34–35
15:         $\mathcal{L}_{\text{Total}} \leftarrow \mathcal{L}_{\text{data}} + \mathcal{L}_{\text{PDE-UV}}$
16:         $\mathcal{O}.\text{zero\_grad}()$
17:         $\mathcal{L}_{\text{Total}}.\text{backward}()$
18:         $\mathcal{O}.\text{step}()$
19:     **end for**
20: **end for**

---

## 3.2 PHYSICAL-INFORMED CHARACTERISTICS OF M²F-PINN

Two partial differential equations (PDEs) are employed in this study, which impose physical constraints on the two variables of $U$ and $V$ respectively, so as to establish the accurate spatiotemporal evolution of ocean current fields. The momentum equations (in the zonal direction and meridional direction) are indispensable for describing ocean circulation. They directly drive thermohaline transport, energy cycles, biological diffusion, and climate mechanisms, and serve as key factors in both dynamic modeling and data-driven forecasting.

**Momentum equation in zonal direction.** The momentum equation is the application of Newton's second law in the ocean, describing the variation of velocity in time and space. The momentum equation in the zonal direction can be expressed as follows:

$$\frac{\partial U}{\partial t} + U\frac{\partial U}{\partial x} + V\frac{\partial U}{\partial y} = \nu\nabla^2 U \tag{4}$$

where $\nu$ is the eddy viscosity coefficient. This equation reflects that the variation of flow velocity is jointly influenced by the inertial term, Coriolis force, and gravitational gradient.

**Momentum equation in meridional direction.** Similarly, the momentum equation in the meridional direction is expressed as:

$$\frac{\partial V}{\partial t} + U\frac{\partial V}{\partial x} + V\frac{\partial V}{\partial y} = \nu\nabla^2 V \tag{5}$$

These two sets of momentum equations constitute the fundamental equations of ocean dynamics.

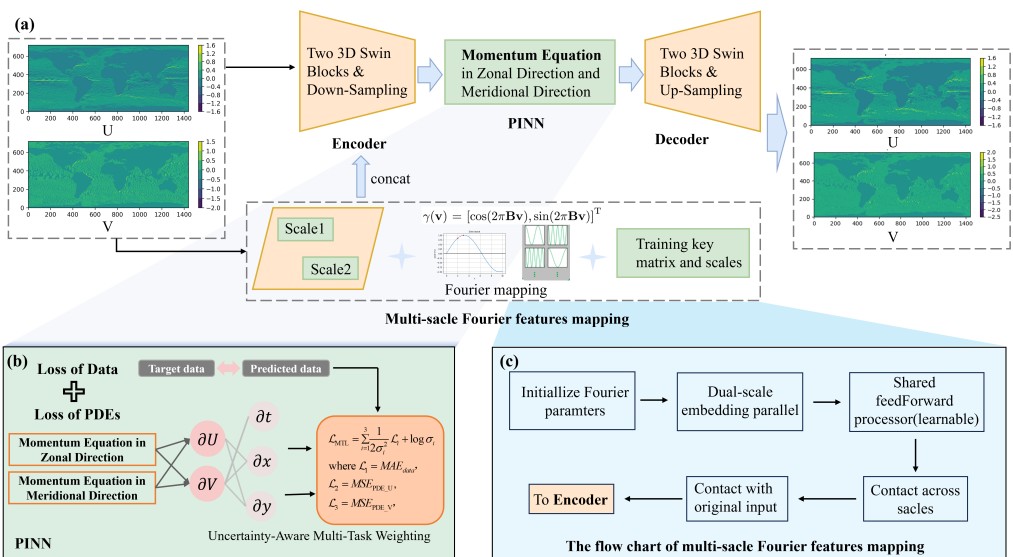

Figure 1: **Overview of the M$^2$F-PINN framework.** First, ocean current fields are fed into a multi-scale Fourier feature mapping module to obtain learned representations. Second, the reconstructed fields are constrained by the two momentum equations from geophysical fluid dynamics. Finally, the multi-scale Fourier features mapping captures both low- and high-frequency components of the ocean currents.

### 3.3 MULTI-SCALE FOURIER REPRESENTATION LEARNING OF M$^2$F-PINN

To mitigate spectral bias and expose both low- and high-frequency oceanic structures to the network, we embed space–time coordinates with a multi-scale Fourier map. This section formalizes the mapping, analyzes its NTK behavior, and explains how it interacts with the physics residuals.

#### 3.3.1 NTK VIEW: HOW FOURIER BANDS RESHAPE LEARNING RATES

By Equation 3, the error dynamics satisfy $\dot{e}(t) = -K\,e(t)$, where $t$ denotes time in error dynamics and $e(t)$ represents the error vector. With the multi-scale Fourier embedding $\Gamma(\cdot)$, we simply replace the kernel by $K_\Gamma$ (entries defined as in Equation 3 but with features $\Gamma$). We then focus on the Fourier-specific spectrum: on near-uniform grids with periodic boundaries, $K_\Gamma$ is approximately translation-invariant; its eigenvectors are (discrete) Fourier modes $\{\varphi_k\}$, and the eigenvalues equal the discrete Fourier transform of the kernel's first row. For the Gaussian kernel,

$$\lambda_k(\sigma) \propto \exp\left(-\frac{\|\omega_k\|^2}{2\sigma^2}\right), \tag{6}$$

where $\omega_k$ denotes the frequency mode. So increasing $\sigma$ flattens the spectrum and raises the learning rates of high-frequency modes. With multiple bands, the effective spectrum becomes $\bar{\lambda}_k = \sum_\ell \alpha_\ell^2 \lambda_k(\sigma_\ell)$. A 1-D calculation and the dependence on grid spacing are given in Appendix A.2.

#### 3.3.2 EXTENSION TO PINNS: COUPLING DATA AND PHYSICS THROUGH A BLOCK NTK

In M$^2$F-PINN, we optimize a composite objective

$$\mathcal{L} = \underbrace{\mathcal{L}_{\text{data}}}_{\text{observations}} + \lambda_{\text{pde}} \underbrace{\mathcal{L}_{\text{pde}}}_{\text{physics residual}}. \tag{7}$$

Stack the errors evaluated on observation sites and PDE collocation points as $E(t) = [e_u(t); e_r(t)]$. Linearizing the dynamics yields the block NTK

$$\dot{E}(t) = -K(t) E(t), \quad K(t) = \begin{bmatrix} K_{uu}(t) & K_{ur}(t) \\ K_{ru}(t) & K_{rr}(t) \end{bmatrix} \tag{8}$$

where $K_{uu}$ is the data NTK, $K_{rr}$ is the residual NTK, and $K_{ur} = K_{ru}^{\top}$ are cross terms induced by parameter sharing. More details in Appendix A.3.

## 4 RESULTS

### 4.1 EXPERIMENTAL SETUP

**Dataset.** The GLORYS12 high-resolution reanalysis data provided by the Copernicus Marine Service Jean-Michel et al. (2021) is used to train and evaluate of the model. From this dataset, $U$ and $V$ are adopted as input variables. To rigorously assess generalization, the dataset is split into two subsets: a training set (2005) and a test set (2006–2008).

**Experiments details.** We conduct experiments over the global ocean domain using a continuous four-year observation period on the GLORYS12 reanalysis dataset. In the performance, ablation, and robust experiments, the depths of coordinate axis in used dataset is distributed from 0.49 m to 130.7 m. Specifically, the depths includes the following 13 depths: 0.49 m, 2.65 m, 5.08 m, 7.93 m, 11.40 m, 15.81 m, 21.60 m, 29.44 m, 40.34 m, 55.76 m, 77.85 m, 109.73 m, 130.67 m. The experiments are conducted on a computer equipped with an Intel Xeon Platinumn 8352, 128GB of RAM, and two NVIDIA A100 80GB GPUs for model training and testing. We adopt the Adam optimizer with an initial learning rate of $5 \times 10^{-4}$, which decays following a cosine annealing schedule. The weight decay is set to $3 \times 10^{-6}$, and the model is trained for 100 epochs.

The evaluation metrics include Root Mean Square Error (RMSE), Anomaly Correlation Coefficient (ACC), and Physical Inconsistency Coefficient (PIC). The definitions of the first two are provided in the Appendix A.6, while the definition of PIC is as follows:

$$\text{PIC}(v, t) = \frac{1}{n} \sum_{i=1}^{n} (f(v_{target}) - f(v_{pred}))^2 \tag{9}$$

where $f$ denotes the residual of the physical loss in Equation 2 $\sim$ 3, $v_{target}$ represents the ground truth of the oceanographic variable at the next time step, and $v_{pred}$ is the predicted value obtained from the model based on the previous time step.

### 4.2 EXPERIMENT PERFORMANCE

The experimental data include $U$ and $V$ variables of ocean currents across the global ocean region. For the forecasting accuracy of multi-variables in the ocean, we selected deep learning baselines including Convolutional Neural Networks (CNN, specifically ResNet) and Recurrent Neural Networks (RNN, specifically LSTM), as well as three additional baselines: MeshGraphNets (based on Graph Neural Networks), Fourier Neural Operators (FNO), and Hard-Constrained PINN (HardC) Négiar et al. (2022). Meanwhile, the physics-informed model, M²F-PINN, was integrated into the aforementioned baselines to form M²F-CNN and M²F-RNN. In addition, two excellent ocean forecasting models, XiHe and WenHai, are included for comparison. All models are trained on data form 2005 and evaluated on unseen data from 2006-2008, and three random trials are additionally run. Reported metrics represent average performance over the 2006-2008 test period.

As shown in Table 1, the proposed M²F-PINN model almost achieves the best performance in terms of the RMSE, ACC, and PIC metrics. For the U and V variables, M²F-PINN achieves the RMSE of 0.03, with accuracy values reaching 0.972 and 0.953, respectively, while its PIC values are as low as 0.71 and 0.80. This performance outperforms traditional deep learning baselines and their variants, as well as the ocean-specific models XiHe and WenHai. M²F-PINN imposes constraints on ocean dynamics through multi-physics-informed mechanism, and effectively learns high-frequency and low-frequency data information via multi-scale Fourier feature mapping—thereby enhancing the prediction accuracy of ocean current fields. Figure 2 and Figure 3 present global visualizations of the $U$ and $V$ generated by the M²F-PINN model for four consecutive days in September, 2006.

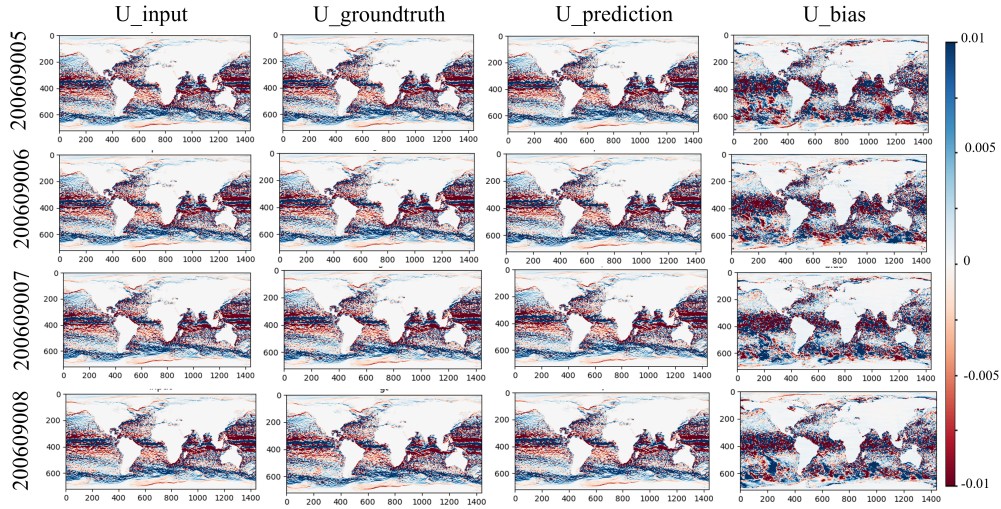

Figure 2: Visualization of $U$ variable over four consecutive days. Columns from left to right show: input, ground truth, prediction, and bias; each row represents one of four consecutive days.

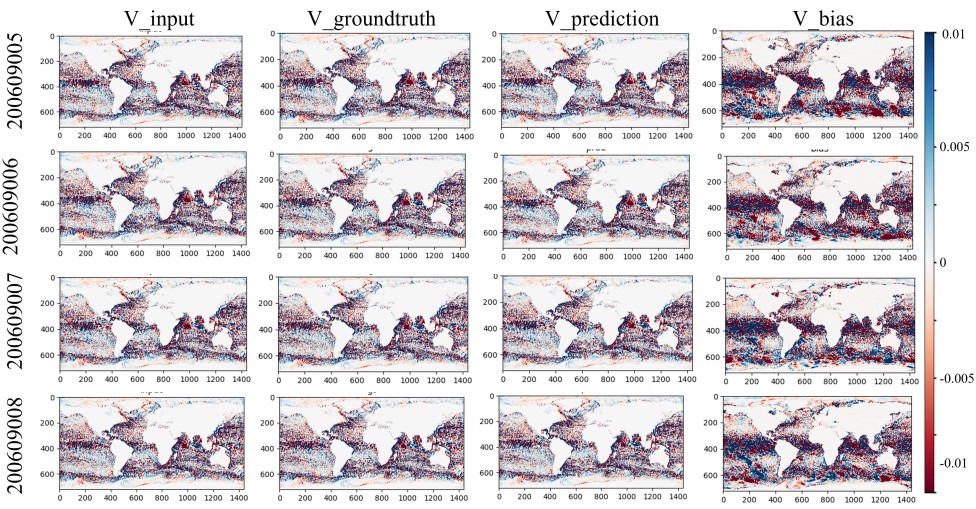

Figure 3: Visualization of $V$ variable over four consecutive days.

Beyond the 1-day forecasting horizon, we also test the model performance on medium-range ocean forecasting (1-day, and 7-day) as well as long-term and seasonal ocean forecasting (30-day, and 60-day). Multi-day forecasting is achieved through direct multi-step regression. Specifically, we independently train forecasting models for 1-day, 7-day, 30-day, and 60-day lead times. As shown in Table 2, the ocean forecasts across different prediction horizons are presented, demonstrating the excellent generalization ability of the M$^2$F-PINN model in long-term prediction. The 1-day setting is best across all metrics (e.g., $U/V$-RMSE 0.03/0.03, $U/V$-ACC 0.972/0.953, $U/V$-PIC 0.71/0.80). As the horizon lengthens, errors increase and accuracies decrease in a smooth trend—by 60 days, $U/V$-RMSE reaches 0.19/0.17 and $U/V$-ACC 0.752/0.704, with $U/V$-PIC rising to 2.25/1.90. The reported uncertainties are small (mostly $10^{-3}$ scale), indicating stable performance across runs. For additional multi-day experimental results comparing different methods, see the appendix.

Table 1: Experimental performance comparison.

| Models | $U$-RMSE($\downarrow$) | $V$-RMSE($\downarrow$) | $U$-ACC($\uparrow$) | $V$-ACC($\uparrow$) | $U$-PIC($\downarrow$) | $V$-PIC($\downarrow$) |
|---|---|---|---|---|---|---|
| CNN | $5.36 \pm 0.1781$ | $19.55 \pm 0.1917$ | $0.942 \pm 0.0023$ | $0.742 \pm 0.0026$ | $7250.50 \pm 388.35$ | $9802.44 \pm 272.30$ |
| M²F-CNN | $0.05 \pm 0.0079$ | $0.06 \pm 0.0129$ | $0.955 \pm 0.044$ | $0.926 \pm 0.0053$ | $4610.53 \pm 0.4166$ | $5378.97 \pm 0.2349$ |
| RNN | $10.16 \pm 8.3810$ | $24.24 \pm 8.2873$ | $0.862 \pm 0.1386$ | $0.584 \pm 0.2757$ | $9686.31 \pm 0.0632$ | $11917.8 \pm 0.4382$ |
| M²F-RNN | $0.12 \pm 0.0144$ | $0.12 \pm 0.0133$ | $0.746 \pm 0.0015$ | $0.595 \pm 0.0045$ | $7063.19 \pm 639.37$ | $9562.37 \pm 449.80$ |
| MeshGraphNets | $0.04 \pm 0.0047$ | $0.04 \pm 0.0060$ | $0.938 \pm 0.0031$ | $0.920 \pm 0.0025$ | $8004.71 \pm 0.6538$ | $11558.25 \pm 0.5353$ |
| FNO | $0.14 \pm 0.0049$ | $0.11 \pm 0.0052$ | $0.035 \pm 0.0045$ | $-0.030 \pm 0.0025$ | $7237.35 \pm 1.0132$ | $10686.15 \pm 0.9500$ |
| HardC | $0.07 \pm 0.0038$ | $0.06 \pm 0.0024$ | $0.661 \pm 0.0087$ | $0.665 \pm 0.0073$ | $1.40 \pm 0.0824$ | $1.876 \pm 0.0730$ |
| XiHe | $0.19$ | $0.19$ | $0.963$ | $0.938$ | $380907.58$ | $441561.92$ |
| WenHai | $0.17$ | $0.16$ | $0.903$ | $0.923$ | $1344.1$ | $84768.29$ |
| **M²F-PINN** | $\mathbf{0.03 \pm 0.0003}$ | $\mathbf{0.03 \pm 0.0001}$ | $\mathbf{0.972 \pm 0.008}$ | $\mathbf{0.953 \pm 0.0036}$ | $\mathbf{0.71 \pm 0.0032}$ | $\mathbf{0.80 \pm 0.0063}$ |

Table 2: Forecasting performance across different prediction horizons.

| Models | $U$-RMSE($\downarrow$) | $V$-RMSE($\downarrow$) | $U$-ACC($\uparrow$) | $V$-ACC($\uparrow$) | $U$-PIC($\downarrow$) | $V$-PIC($\downarrow$) |
|---|---|---|---|---|---|---|
| M²F-PINN-1day | $\mathbf{0.03 \pm 0.0003}$ | $\mathbf{0.03 \pm 0.0001}$ | $\mathbf{0.972 \pm 0.008}$ | $\mathbf{0.953 \pm 0.0036}$ | $\mathbf{0.71 \pm 0.0032}$ | $\mathbf{0.80 \pm 0.0063}$ |
| M²F-PINN-7day | $0.13 \pm 0.0024$ | $0.14 \pm 0.0053$ | $0.841 \pm 0.0081$ | $0.720 \pm 0.0051$ | $1.61 \pm 0.0082$ | $1.27 \pm 0.0068$ |
| M²F-PINN-30day | $0.18 \pm 0.0031$ | $0.16 \pm 0.0046$ | $0.690 \pm 0.0018$ | $0.618 \pm 0.0125$ | $1.11 \pm 0.0224$ | $1.50 \pm 0.0124$ |
| M²F-PINN-60day | $0.19 \pm 0.0006$ | $0.17 \pm 0.0007$ | $0.752 \pm 0.0110$ | $0.704 \pm 0.00758$ | $2.25 \pm 0.0042$ | $1.90 \pm 0.0103$ |

### 4.3 ABLATION STUDY

In the ablation study, the core components of the M²F-PINN method are ablated: the PINN-base variant removes the Fourier module, while the Data-base variant removes both the PINN module and the Fourier module. As shown in Table 3, the M²F-PINN model consistently achieves near-optimal performance across all evaluation metrics, including the RMSE, ACC, and PIC of both the $U$ and $V$ flow fields. On PINN-Base model, the RMSE slightly increases from 0.03 to 0.04, while the ACC decreases by approximately 0.1. However, the most pronounced change is observed in the PIC metric, indicating that the multi-scale Fourier network enhances the PINN's ability to capture the underlying physical laws. Similarly, in the data-based experiments, both the RMSE and ACC exhibit minor decreases, whereas the PIC again shows the largest decline, further demonstrating that the incorporation of PINN facilitates the learning of more interpretable physical patterns.

### 4.4 EXPLAINABILITY FROM PIC OF M²F-PINN

The PIC values quantify the deviation between model predictions and the true underlying ocean dynamics. Figures 4 and 5 visualize the PIC values of the $U$ and $V$ variables, respectively, across the six models used in the ablation experiments. It can be observed that the differences between Figure 4(a) of M²F-PINN and Figure 4(b)(c)(d) are minimal, whereas the first four panels exhibit a clear global advantage over the PINN-base and Data-base models. This result indicates that M²F-PINN better conforms to ocean dynamical principles and achieves superior physical consistency.

### 4.5 POWER-SPECTRUM ANALYSIS OF M²F-PINN

As illustrated in Figure 6, multi-scale power spectrum comparison demonstrating the M²F-PINN model's capability to capture ocean dynamics from fine-scale (10-100 km) to large-scale (>500 km) features. M²F-PINN accurately captures both meso-scale (100-500 km) and large-scale oceanic variability. The predicted spectrum exhibits near-perfect overlap with the ground truth at these scales, demonstrating the M²F-PINN model's strong ability to recover multi-scale ocean dynamics. In contrast, the PINN variant fails to capture these multi-scale structures.

## 5 RELATED WORKS

**Neural Network-Based Weather and Ocean Forecasting.** Numerous studies have been conducted on neural network-based weather forecasting. FourCastNet Pathak et al. (2022) leverages an adap-

Table 3: Ablation study of M$^2$F-PINN.

| Description | $U$-RMSE($\downarrow$) | $V$-RMSE($\downarrow$) | $U$-ACC($\uparrow$) | $V$-ACC($\uparrow$) | $U$-PIC($\downarrow$) | $V$-PIC($\downarrow$) |
|---|---|---|---|---|---|---|
| **M$^2$F-PINN** | **0.03** | **0.03** | **0.972** | **0.953** | **0.71** | **0.80** |
| PINN-Base | 0.04 | 0.04 | 0.965 | 0.941 | 86.94 | 57.79 |
| Data-Base | 0.05 | 0.05 | 0.951 | 0.914 | 6127.38 | 7434.28 |

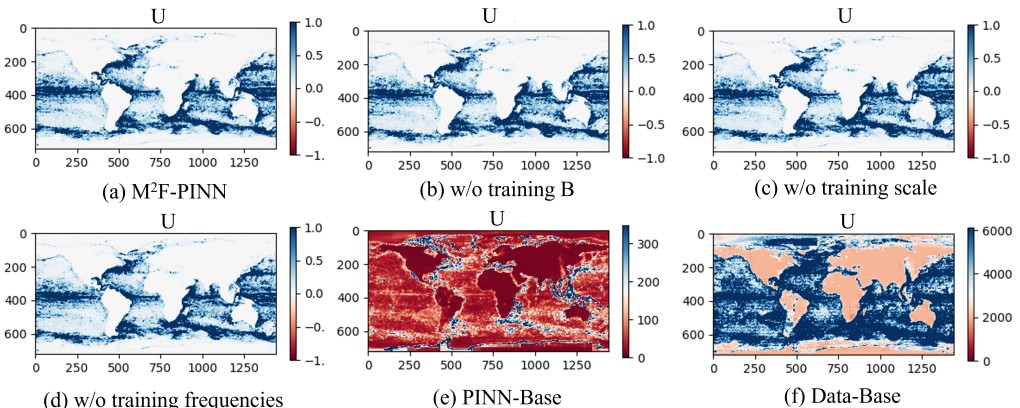

(a) M$^2$F-PINN     (b) w/o training B     (c) w/o training scale

(d) w/o training frequencies     (e) PINN-Base     (f) Data-Base

Figure 4: Visualization of $U$ PIC values.

tive Fourier neural operator network, while Pangu Weather Bi et al. (2023) introduces a hierarchical temporal aggregation method to minimize the iterative loss accuracy of autoregressive predictions at different time steps.The Fengwu series Chen et al. (2023a); Han et al. (2024) models serve as comprehensive climate forecasting models covering multiple scales, including nowcasting, medium-range forecasting, and multi-year to interannual forecasting. The Fuxi series Chen et al. (2023b); Zhong et al. (2024b;a); Chen et al. (2024); Zhong et al. (2024c) models adopt a cascaded architecture optimized for different forecasting horizons and represent the first operationally viable high-precision global weather cycling assimilation and forecasting AI system that integrates real observations into forecasting. Additionally, GraphCast Lam et al. (2023), GenCast Price et al. (2023), and OneForecast Gao et al. (2025) employ graph neural networks to model the Earth's atmospheric state on a spherical grid. In contrast, ocean forecasting still requires further development. AI-GOMS Xiong et al. (2023) adopts a Fourier-based masked autoencoder as its backbone structure; XiHe Wang et al. (2024), a hierarchical transformer, effectively captures both local and global oceanic information; Kunpeng Zhao et al. (2025) implements a longitude-cyclic deformable convolutional network to achieve fine-grained modeling of multi-scale oceanic features; and Langya Yang et al. (2024) develops a cross-spatiotemporal and atmosphere-forced ocean forecasting system guided by physics-informed loss functions. However, the aforementioned ocean forecasting models exhibit insufficient utilization of frequency-domain and high-frequency oceanic information, and face challenges in terms of generalization and physical interpretability.

**PINNs in Scientific Computing and Climate-Ocean Modeling.** The emergence of PINNs has addressed the most critical limitation of neural networks—their nature as black-box systems lacking interpretability and physical consistency. Raissi et al. Raissi et al. (2020) demonstrated the effectiveness of PINNs in solving classical nonlinear PDE problems across various interdisciplinary fields. Subsequently, PINNs have been widely applied in materials science, mechanics, fluid dynamics, and other scientific and engineering domains Diligenti et al. (2017); Liu et al. (2022); Zhang et al. (2022); Abueidda et al. (2023). In the frequency domain, Xu et al. Xu et al. (2019) utilized Random Fourier Features Rahimi & Recht (2007) to approximate stationary kernels with sinusoidal input mappings, and proposed techniques for adjusting mapping parameters. Tancik et al. Tancik et al. (2020) employed Fourier feature mapping to convert the effective Neural NTK into a stationary kernel with adjustable bandwidth. Wang et al. Wang et al. (2021) constructed a novel architecture

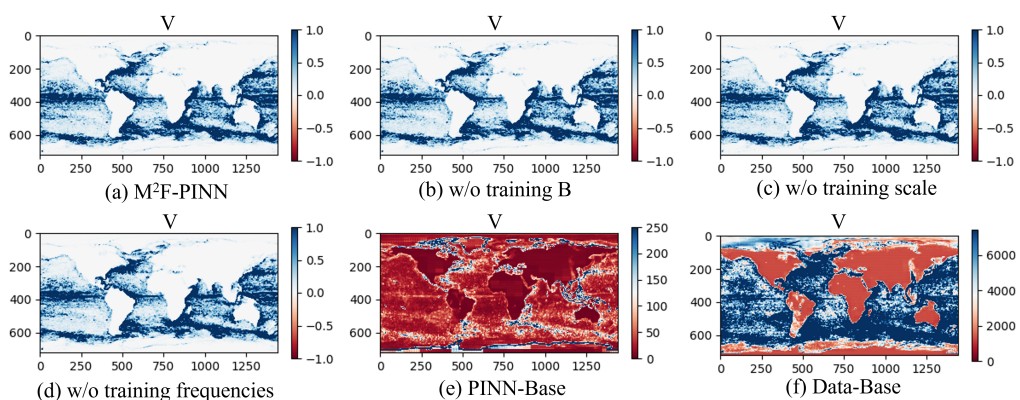

Figure 5: Visualization of $V$ PIC values.

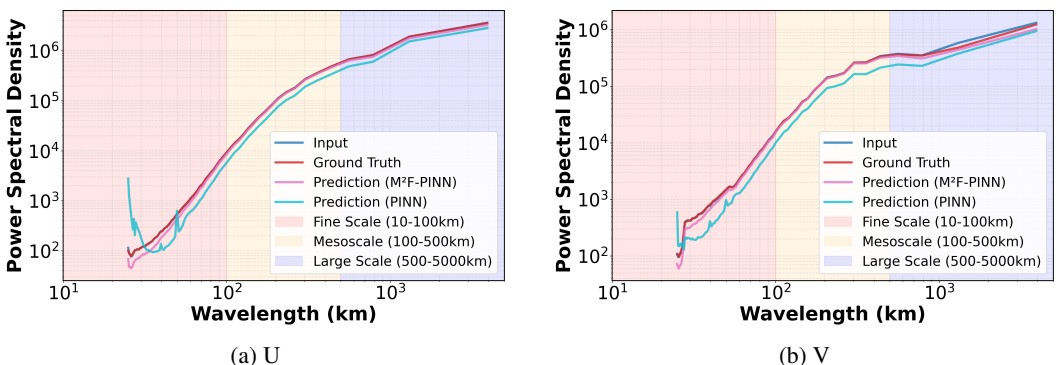

Figure 6: Power-spectrum analysis of M$^2$F-PINN and PINN.

incorporating spatiotemporal and multi-scale random Fourier features, and verified how such a co-ordinate embedding layer could yield robust and accurate PINN models. Currently, research on the application of PINNs in climate and ocean modeling includes the following: ClimODE Verma et al. (2024) and PIHC-MoE Chalapathi et al. (2024) integrate implicit PDE constraints into the model architecture; NeuralGCM Kochkov et al. (2024) parameterizes atmospheric dynamics; and WenHai Cui et al. (2025) incorporates physical parameterization of air-sea coupling into deep neural networks. However, studies on introducing Fourier mapping networks into ocean forecasting remain relatively scarce. Nevertheless, high-frequency information of ocean flow fields is equally crucial for prediction accuracy and thus requires consideration.

## 6 FURTHER DISCUSSION AND CONCLUSION

In this paper, we introduces a novel multi-scale frequency-domain PINN forecasting algorithm, M²F-PINN. This algorithm leverages the 3D Swin Transformer for autoregressive learning of data patterns, while utilizing Fourier representation learning in the multi-scale frequency domain to effectively alleviate the spectral bias issue. Additionally, it incorporates physically interpretable physical knowledge through the physical constraints of multi-PINN.

**Limitations and Future Work.** PINNs incorporate the momentum equation under the Navier-Stokes equations, while the coupled constraints of multi-variables in ocean systems—such as salinity and temperature—have not been fully integrated. Another potential research direction is that the predictive capability of PINNs for more extreme ocean events (e.g., tsunamis and severe storms) still requires further validation.

ETHICS STATEMENT

The contributions of this work advance predictive algorithms in the field of deep learning. This algorithm may have an impact on several downstream applications. While we hope that it will not lead to any adverse consequences, as with any predictive tool, there remains a possibility of misuse.

REPRODUCIBILITY STATEMENT

Our code is provided in the Supplementary Material.

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

# A APPENDIX

## A.1 PROOF OF SPECTRAL BIAS THROUGH THE LENS OF THE NEURAL TANGENT KERNEL

**Assumptions.** We work in the standard infinite-width (or lazy-training) regime with gradient-flow dynamics and small learning rates. When Fourier features are used on a near-uniform training grid with periodic boundary conditions, the NTK is approximately translation-invariant and its eigenvectors align with discrete Fourier modes. The subsequent analysis relies on these assumptions.

Fourier feature embedding is a technique for transforming input features, enabling the network to learn multi-scale variations in the output. The core idea of Fourier feature embedding is to map the input $v$ to an embedding $\gamma(\mathbf{v})$, defined by Equation 28. We consider the neural network as a fully connected network with scalar outputs, where parameters $\theta$ are initialised from a Gaussian distribution $\mathcal{N}(0,1)$. The training dataset is denoted as $\{X_{\text{train}}, Y_{\text{train}}\}$, where $X_{\text{train}} = (x_i)_{i=1}^N$ and $Y_{\text{train}} = (y_i)_{i=1}^N$. The loss function is defined as minimising the mean squared error:

$$L(\theta) = \frac{1}{N} \sum_{i=1}^N |f(x_i, \theta) - y_i|^2 \tag{10}$$

NTK is defined as:

$$K_{ij} = K(x_i, x_j) = \left\langle \frac{\partial f(x_i, \theta)}{\partial \theta}, \frac{\partial f(x_j, \theta)}{\partial \theta} \right\rangle \tag{11}$$

Where, NTK is a kernel matrix measuring the similarity between inputs $x_i$ and $x_j$, defined through the gradient inner product of the network's parameters. Under infinitely wide networks and small learning rates, NTK converges to a deterministic kernel $K^*$ and remains invariant during training Jacot et al. (2018).

$$\frac{df(X_{\text{train}}, \theta(t))}{dt} \approx -K \cdot (f(X_{\text{train}}, \theta(t)) - Y_{\text{train}}) \tag{12}$$

Equation 12 is derived from the continuous limit of gradient descent (gradient flow). Assuming an infinitesimal learning rate, the change in the network output f obeys a linear ODE. This models network training as a linear system, with NTK K governing the dynamics.

$$f(X, \theta(t)) = Y_{\text{train}} - e^{-Kt}(Y_{\text{train}} - f(X, \theta(0))) \tag{13}$$

Solving the ODE in Equation 12 with the initial condition $f(X_{\text{train}}, \theta(0)) \approx 0$ (as induced by random initialization) yields Equation 13. The network output approaches the target $Y_{\text{train}}$ as $t$ increases, with the convergence rate governed by the matrix exponential $e^{-Kt}$ (i.e., by the spectrum of $K$).

The kernal K-function of NTK is defined as: $K = Q^T \Lambda Q$, where $Q$ is an orthogonal matrix (whose columns are the eigenvectors $q_i$), and $\Lambda$ is a diagonal matrix (whose diagonal entries are the eigenvalues $\lambda_i$). The decomposition of error under the basis of features is

$$Q^T(f(X_{\text{train}}, \theta(t)) - Y_{\text{train}}) = -e^{-\Lambda t} Q^T Y_{\text{train}} \tag{14}$$

where $\Lambda$ denotes the eigenvalue matrix, $Q$ denotes the eigenvector matrix, $q_i$ denotes the eigenvector, and $\lambda_N$ denotes the eigenvalue. Further derivation yields:

$$\begin{bmatrix} q_1^T \\ q_2^T \\ \vdots \\ q_N^T \end{bmatrix} (f - Y_{\text{train}}) = \begin{bmatrix} e^{-\lambda_1 t} & & & \\ & e^{-\lambda_2 t} & & \\ & & \ddots & \\ & & & e^{-\lambda_N t} \end{bmatrix} \begin{bmatrix} q_1^T \\ q_2^T \\ \vdots \\ q_N^T \end{bmatrix} Y_{\text{train}} \tag{15}$$

Specifically, substitute the spectral decomposition $e^{-Kt} = Q^T e^{-\Lambda t} Q$ and then multiply by $Q^T$ on the left. The error component along the $i$th eigenvector decays as $e^{-\lambda_i t}$. Large $\lambda_i$ values correspond to rapid convergence, while small values indicate slow convergence. This explains 'spectral bias': if high-frequency eigenvectors correspond to small $\lambda_i$, the network favours learning low-frequency components Ronen et al. (2019); Rahaman et al. (2019). Furthermore, total Error Decomposition is:

$$f - Y_{\text{train}} = \sum_{i=1}^N (f - Y_{\text{train}}, q_i) q_i = \sum_{i=1}^N (e^{-\lambda_i t} q_i^T Y_{\text{train}}) q_i \tag{16}$$

The training error is projected onto the NTK feature basis, with the network first learning the directions of large eigenvalues (typically low-frequency components). The above derivation emphasises that spectral deviation is fundamentally a "eigenvector deviation", as eigenvectors determine learning frequency while eigenvalues determine velocity.

## A.2 Proof of large-scale $\sigma$ leads to high-frequency eigenvectors

**Domain & measure.** All integral operators below are defined on the one-dimensional torus $\mathbb{T} = [0,1]$ (or $[0,2\pi]$) with the uniform measure; on $[0,1]$ we use the periodic extension of the kernel. This ensures that translation-invariant kernels admit Fourier eigenfunctions.

Consider a two-layer unbiased neural network with Fourier characteristics, shown in Equation 1. According to Equation 11, the NTK is:

$$K(x_i, x_j) = \frac{1}{m} \sum_{k=1}^{m} \cos(b_k^T(x_i - x_j)) \tag{17}$$

To investigate the characteristic system of the kernel function $K$, we consider the limit case of $K$ as the number of points approaches infinity. Under this limit condition, the characteristic system of $K$ approaches that of the kernel function $K(x, x')$ satisfying the following equation.

$$\int_C K(x, x')g(x') \, dx' = \lambda g(x) \tag{18}$$

As the number of points approaches infinity, the eigenvectors of the NTK matrix converge to the eigenfunctions of the integral operator.

$$K(x, x') = \frac{1}{m} \sum_{k=1}^{m} \cos\left(z_k^T(x - x')\right) \tag{19}$$

To better understand the behaviour of eigenfunctions and their corresponding eigenvalues, we consider a simpler case by setting $n = 1$ and $m = 1$. Specifically, we take the input $x \in \mathbb{R}$, a compact domain $C = [0,1]$, and Fourier characteristics $B = b \in \mathbb{R}$ sampled from the Gaussian distribution $N(0, \sigma^2)$. The kernel function can then be expressed as

$$K(x, x') = \cos(b(x - x')) \tag{20}$$

Under these circumstances, we may compute precise expressions for the eigenfunctions and their corresponding eigenvalues. For the kernel function K, its non-zero eigenvalues are given by the following formula:

$$\lambda = \frac{1}{2} \pm \frac{\sin b}{2b} \tag{21}$$

For $K(x, x') = \cos(b(x - x'))$ on $[0,1]$, the integral operator has rank at most 2 with eigenvalues above; for an interval of length $L$, $\lambda_{\pm}(b) = \frac{L}{2} \pm \frac{\sin(bL)}{2b}$. The corresponding eigenfunction g(x) must take the following form:

$$g(x) = C_1 \cos(bx) + C_2 \sin(bx) \tag{22}$$

where $C_1$ and $C_1$ are constants. We immediately observe that the frequency of the eigenfunctions is determined by the parameter $b$, whilst the spacing between eigenvalues is governed by $sinb/b$. Furthermore, it should be noted that the parameter $b$ is randomly sampled from a Gaussian distribution $N(0, \sigma^2)$, implying that the larger the chosen value of $\sigma$, the higher the probability of $b$ assuming larger numerical values. Consequently, we may conclude that 'spectral bias' effectively corresponds to 'eigenvector bias'—namely, the principal eigenvectors associated with larger eigenvalues determine the frequency range the network prioritises for learning. Within this simplified model, larger $\sigma$ values not only induce higher frequencies in the feature functions but also narrow the intervals between eigenvalues. In network performance, selecting an appropriate $\sigma$ value to align the frequency of NTK principal eigenvectors with that of the objective function plays a crucial role. This not only accelerates convergence but also effectively enhances network performance. Consequently, Fourier feature analysis not only effectively addresses spectral bias but also accelerates convergence of the high-frequency components of the objective function.

**Proposition (Spectrum of the expected RBF kernel on $\mathbb{T}^d$).** With random Fourier features $B \sim \mathcal{N}(0, \sigma^2 I)$, the expected kernel

$$K_\sigma(\Delta) = \mathbb{E}[\cos(B^\top \Delta)] = \exp\left(-\frac{\sigma^2}{2}\|\Delta\|^2\right) \tag{23}$$

has eigenfunctions $\varphi_k(x) = e^{i2\pi k \cdot x}$ and eigenvalues equal to the discrete Fourier coefficients:

$$\lambda_k(\sigma) \propto \widehat{K_\sigma}(2\pi k) \propto \exp\left(-2\pi^2\|k\|^2/\sigma^{-2}\right) \tag{24}$$

Hence, as $\sigma$ increases, the spectrum flattens and high-frequency modes receive larger weights; with multiple bands, $\bar{\lambda}_k = \sum_\ell \alpha_\ell^2 \lambda_k(\sigma_\ell)$.

### A.3 PROOF OF THE NTK FEATURE SYSTEM DETERMINES PINNs TRAINING

We now turn our attention back to physical information neural networks for addressing forward and inverse problems involving partial differential equations, whose solutions may exhibit multi-scale behaviour. The NTK employed by PINNs exhibits a slightly more intricate network architecture than that utilised by conventional regression models. To this end, we adopt the experimental framework proposed by Wang et al. Wang et al. (2022), selecting generalised partial differential equations with appropriate boundary conditions and employing the corresponding training datasets $\left\{\left(x_b^i, g\left(x_z^i\right)\right)\right\}_{i=1}^{N_z}, \left\{\left(x_r^i, f\left(x_r^i\right)\right)\right\}_{i=1}^{N_r}$.

Based on these assumptions, we define the neural network gradient computation for PINNs as follows:

$$\boldsymbol{K}(t) = \begin{bmatrix} K_{uu}(t) & K_{uv}(t) \\ K_{vu}(t) & K_{vv}(t) \end{bmatrix} \tag{25}$$

where $K_{vu}(t) = K_{uu}^T(t)$ $K_{uu}(t) \in \mathbb{R}^{N_z \times N_z}$ $K_{uu}(t) \in \mathbb{R}^{N_z \times N_v}$ $K_{vv}(t) \in \mathbb{R}^{N_v \times N_v}$, its $(i,j)_{th}$ element is given by the following formula:

$$(K_{uu})_{ij}(t) = \left\langle \frac{d\mathcal{B}[u](x_z^i, \theta(t))}{d\theta}, \frac{d\mathcal{B}[u](x_z^j, \theta(t))}{d\theta} \right\rangle \tag{26}$$

$$(K_{uv})_{ij}(t) = \left\langle \frac{d\mathcal{B}[u](x_z^i, \theta(t))}{d\theta}, \frac{d\mathcal{N}[u](x_v^j, \theta(t))}{d\theta} \right\rangle \tag{27}$$

$$(K_{vv})_{ij}(t) = \left\langle \frac{d\mathcal{N}[u](x_v^i, \theta(t))}{d\theta}, \frac{d\mathcal{N}[u](x_v^j, \theta(t))}{d\theta} \right\rangle \tag{28}$$

Subsequently, the training dynamics of PINNs under gradient descent with an infinitesimal learning rate can be characterised by the following system of ordinary differential equations.

$$\begin{bmatrix} \frac{d\mathcal{B}[\boldsymbol{u}](\boldsymbol{x}_z, \boldsymbol{\theta}(t))}{dt} \\ \frac{d\mathcal{N}[\boldsymbol{u}](\boldsymbol{x}_v, \boldsymbol{\theta}(t))}{dt} \end{bmatrix} = -\begin{bmatrix} \boldsymbol{K}_{uu}(t) & \boldsymbol{K}_{uv}(t) \\ \boldsymbol{K}_{vu}(t) & \boldsymbol{K}_{vv}(t) \end{bmatrix} \cdot \begin{bmatrix} \mathcal{B}[\boldsymbol{u}](\boldsymbol{x}_z, \boldsymbol{\theta}(t)) - \boldsymbol{g}(\boldsymbol{x}_z) \\ \mathcal{N}[\boldsymbol{u}](\boldsymbol{x}_v, \boldsymbol{\theta}(t)) - \boldsymbol{f}(\boldsymbol{x}_v) \end{bmatrix} \tag{29}$$

Then, the NTK framework enables us to demonstrate the following proposition. Assuming the training dynamics of PINNs satisfy the aforementioned equations, and the spectral decomposition of $K_{uu}(0)$ and $K_{vv}(0)$ is given by

$$\begin{aligned} \boldsymbol{K}_{uu}(0) &= \boldsymbol{M}_u^T \Lambda_u \boldsymbol{M}_u^T \\ \boldsymbol{K}_{vv}(0) &= \boldsymbol{M}_v^T \Lambda_v \boldsymbol{M}_v^T \end{aligned} \tag{30}$$

Among these, $M_u$ and $M_v$ are orthogonal matrices formed by the eigenvectors of $K_{uu}(0)$ and $K_{vv}(0)$ respectively, whilst $\Lambda_u$ and $\Lambda_v$ are diagonal matrices whose elements correspond to the eigenvalues of $K_{uu}(0)$ and $K_{vv}(0)$ respectively. Under the given assumptions,(i) For all $t \geq 0$, $K(t) \approx K(0)$; (ii) $K_{uu}(0)$ and $K_{vv}(0)$ are positive definite.

$$\boldsymbol{B} = \boldsymbol{M}_v^T, \boldsymbol{K}_{vu}(0)\boldsymbol{M}_u \tag{31}$$

And obtain,

$$M^T \left( \begin{bmatrix} \mathcal{B}[u](x_b, \theta(t)) \\ \mathcal{N}[u](x_v, \theta(t)) \end{bmatrix} - \begin{bmatrix} g(x_b) \\ f(x_v) \end{bmatrix} \right) \approx e^{-P^T \bar{\Lambda} P t} M^T \begin{bmatrix} g(x_b) \\ f(x_v) \end{bmatrix} \tag{32}$$

where $M = \begin{bmatrix} M_u & 0 \\ 0 & M_v \end{bmatrix}, P = \begin{bmatrix} I & 0 \\ -B\Lambda_u^{-1} & I \end{bmatrix}, \Lambda = \begin{bmatrix} \Lambda_u & 0 \\ 0 & \Lambda_v - B^T \Lambda_u^{-1} B \end{bmatrix}$.

The above argument demonstrates that, under certain assumptions, the NTK feature system of PINNs is determined by the eigenvectors of $K_{uu}$ and $K_{vv}$. This implies that, assuming the NTK matrix is invertible, infinitely wide or sufficiently wide PINNs are equivalent to kernel regression. However, based on the authors' experience, the NTK matrix of PINNs is invariably degenerate. Consequently, in practical applications, one cannot freely perform kernel regression predictions without introducing additional regularisation.

### A.4 LOSS DEFINITION OF M$^2$F-PINN

We embed physics-based constraints into the neural architecture using a set of two PDEs that govern the temporal-spatial evolution of key oceanographic variables of ocean current fields: $U$ and $V$. The M$^2$F-PINN model receives raw ocean state information as input and implicitly learns to estimate their temporal and spatial derivatives. These derivatives are then used to approximate the system's physical evolution over a short time interval $\Delta t$. The overall training objective comprises three loss components: one data-driven prediction loss and two physics-informed residual losses derived from equation 4 $\sim$ 5. The data loss $\mathcal{L}_{\text{data-i}}$, supervises predictions for two oceanographic variables: U and V. The formulation of data loss can be expressed as follows:

$$\mathcal{L}_1 = \sum_{i=1}^{2} \mathcal{L}_{\text{data-i}} \cdot W_i, \quad \text{where } \mathcal{L}_{\text{data-i}} = MSE(v_{\text{pred}} - v_{\text{real}}) \tag{33}$$

Here, $v_{\text{pred}}$ and $v_{\text{real}}$ denote the predicted and ground-truth values of each oceanographic variable, respectively. The weights $W_i$ are fixed and manually calibrated to balance the relative scale of each variable by using the inverse of their early-stage training losses, computed from GLORYS12 reanalysis data spanning 2005–2006. The specific weights are set as follows: [ U: 0.38, V: 0.30].

The remaining two loss terms encode physics-informed constraints derived from PDEs that govern the temporal and spatial evolution of the ocean current. Each physics-informed loss is derived from a corresponding physical law that governs the evolution of oceanographic variables. The second loss $\mathcal{L}_2$ represents momentum equation in zonal direction as shown in Equation 34, the third loss $\mathcal{L}_3$ represents momentum equation in meridional direction as shown in Equation 35. We employ an uncertainty-weighted strategy Kendall et al. (2018) to adaptively adjust the weights of multiple losses.

$$\mathcal{L}_4 = MSE(\frac{\partial U}{\partial t} + U\frac{\partial U}{\partial x} + V\frac{\partial U}{\partial y} - \alpha_{UV}\nabla^2 U) \tag{34}$$

$$\mathcal{L}_5 = MSE(\frac{\partial V}{\partial t} + U\frac{\partial V}{\partial x} + V\frac{\partial V}{\partial y} - \alpha_{UV}\nabla^2 V) \tag{35}$$

The physics-informed losses are formulated independently of the Transformer architecture, ensuring that their computation and optimization are decoupled from the model's neural network components.

### A.5 IMPLEMENTATION DETAILS OF THE MULTI-SCALE FOURIER AND EXPERIMENTS

In the **multi-scale Fourier mapping** network, two Fourier embedding layers (low-frequency and high-frequency) share identical dimensionality. Subsequently, the data passes through a shared feed-forward layer, where the cat layers operate at both high and low dimensions. Finally, the Fourier features are concatenated with the original data before entering the model network. In this paper, we set the initial frequencies to 0.1 Hz and 1.0 Hz. Additionally, we set the B matrix, scale value, and high/low frequency parameters within the Fourier mapping network as variables that the neural

Table 4: More details on M$^2$F-PINN training.

| Description | $U$-RMSE($\downarrow$) | $V$-RMSE($\downarrow$) | $U$-ACC($\uparrow$) | $V$-ACC($\uparrow$) | $U$-PIC($\downarrow$) | $V$-PIC($\downarrow$) |
|---|---|---|---|---|---|---|
| M$^2$F-PINN | **0.030** | **0.030** | **0.972** | 0.953 | 0.71 | **0.80** |
| w/o training B | 0.039 | 0.040 | 0.967 | 0.948 | 0.71 | 0.81 |
| w/o training scale | 0.035 | 0.036 | 0.973 | **0.956** | 0.72 | 0.82 |
| w/o training frequencies | 0.035 | 0.036 | 0.973 | 0.956 | **0.70** | 0.81 |

network can learn, enabling dynamic adaptation to global ocean data. Input dimension is 3 (longitude, latitude, depth), mapping dimension is 16 (feature richness balanced with GPU capacity), frequency factors are 0.1 and 1 (frequency value). Create coordinate network → normalise grid → obtain frequency domain features → transform shape → concatenate Fourier features with original data. Slightly adjust original network parameter values to match Fourier network output shape.

We conducted additional ablation studies on several training-related components, specifically removing the trainable B matrix, the trainable scale, and the trainable frequency parameters. The comparative results are reported in Table 4. As shown, the full M$^2$F-PINN consistently achieves the best or near-best performance across all evaluation metrics, demonstrating the effectiveness and robustness of our complete design.

### A.6 EVALUATION METRICS

When we suppose Given an input variable $V$, the model predicts its future state $\hat{V}$ at the next time step. We evaluate the prediction performance using latitude-weighted Root Mean Squared Error (RMSE) and latitude-weighted Anomaly Correlation Coefficient (ACC). At a specific time step $t$, the RMSE and ACC for predicted oceanographic variables (U, and V) are defined as follows:

$$\text{RMSE}(v, t) = \sqrt{\frac{\sum_{i=1}^{W} \sum_{j=1}^{H} L(i)(\hat{\mathbf{V}}_{i,j,t}^{v} - \mathbf{V}_{i,j,t}^{v})^2}{W \times H}} \tag{36}$$

$$\text{ACC}(v, t) = \frac{\sum_{i,j} L(i) \hat{V}_{i,j,t}^{\prime v} - V_{i,j,t}^{\prime v}}{\sqrt{\sum_{i,j} L(i)(\hat{V}_{i,j,t}^{\prime v})^2 \times \sum_{i,j} L(i)(V_{i,j,t}^{\prime v})^2}} \tag{37}$$

where $L(i)$ is the weight at latitude $\phi_i$. $V'$ denotes the difference between $Y$ and the climatology. In this study, we calculate the annual averages of the above evaluation metrics to assess model performance in a year. This approach aligns with the primary objective of this study—to investigate potential advantages of incorporating physics-informed neural networks into ocean forecasting models.

Beyond predictive accuracy, another key contribution of M$^2$F-PINN is its ability to learn physically consistent predictions. To quantify that, we introduce the Physical Inconsistency Coefficient (PIC) Elabid et al. (2022); Daw et al. (2022), which evaluates the degree to which the model's outputs violate from established physical laws. The PIC is formally defined in Equation 9. Since this operator contains second-order spatial derivatives of the horizontal velocities, PIC naturally reflects discrepancies in the predicted dynamical energy and thus serves as an indicator of physical consistency.

### A.7 A COMPARISON OF TRAINING AND INFERENCE SPEED AGAINST STATE-OF-THE-ART MODELS

The training, inference speed comparison between M$^2$F-PINN and xihe wenhai of the state of the art baseline is as shwon in Table 5. Since neither the XiHe nor WenHai papers report training time, and because only pre-trained weights are publicly available, we were able to evaluate these models only in the inference mode. Consequently, we mark their training time as N/A in our comparison ta-

Table 5: A comparison study of training, inference time of baselines and M$^2$F-PINN.

| Description | Training Time(Hours) | Test Time(Hours) |
|---|---|---|
| CNN | 11 | 0.5 |
| M$^2$F-CNN | 12 | 0.5 |
| RNN | 23 | 0.52 |
| M$^2$F-RNN | 23 | 0.52 |
| MeshGraphNets | 0.25 | 1.0 |
| FNO | 0.33 | 0.27 |
| HardC | 0.8 | 0.77 |
| Xihe | N/A | 3.5 |
| WenHai | N/A | 22[1] |
| **M$^2$F-PINN** | 21 | 0.52 |

Table 6: Forecasting performance across different prediction horizons compare with other methods.

| Models | $U$-RMSE($\downarrow$) | $V$-RMSE($\downarrow$) | $U$-ACC($\uparrow$) | $V$-ACC($\uparrow$) | $U$-PIC($\downarrow$) | $V$-PIC($\downarrow$) |
|---|---|---|---|---|---|---|
| M$^2$F-CNN-1day | $0.05 \pm 0.0079$ | $0.06 \pm 0.0129$ | $0.955 \pm 0.044$ | $0.926 \pm 0.0053$ | $4610.53 \pm 0.4166$ | $5378.97 \pm 0.2349$ |
| M$^2$F-RNN-1day | $0.12 \pm 0.0144$ | $0.12 \pm 0.0133$ | $0.746 \pm 0.0015$ | $0.595 \pm 0.0045$ | $9686.31 \pm 0.0632$ | $11917.8 \pm 0.4382$ |
| **M$^2$F-PINN-1day** | $\mathbf{0.03 \pm 0.0003}$ | $\mathbf{0.03 \pm 0.0001}$ | $\mathbf{0.972 \pm 0.008}$ | $\mathbf{0.953 \pm 0.0036}$ | $\mathbf{0.71 \pm 0.0032}$ | $\mathbf{0.80 \pm 0.0063}$ |
| M$^2$F-CNN-7day | $0.06 \pm 0.0057$ | $0.06 \pm 0.0062$ | $0.928 \pm 0.087$ | $0.889 \pm 0.0041$ | $4873.83 \pm 0.8112$ | $5530.79 \pm 0.6901$ |
| M$^2$F-RNN-7day | $0.34 \pm 0.0080$ | $0.26 \pm 0.0059$ | $0.470 \pm 0.0075$ | $0.498 \pm 0.0064$ | $9688.10 \pm 0.8991$ | $16793.21 \pm 0.6815$ |
| **M$^2$F-PINN-7day** | $\mathbf{0.13 \pm 0.0024}$ | $\mathbf{0.14 \pm 0.0053}$ | $\mathbf{0.841 \pm 0.0081}$ | $\mathbf{0.720 \pm 0.0051}$ | $\mathbf{1.61 \pm 0.0082}$ | $\mathbf{1.27 \pm 0.0068}$ |
| M$^2$F-CNN-30day | $0.11 \pm 0.0120$ | $0.13 \pm 0.0041$ | $0.820 \pm 0.077$ | $0.741 \pm 0.0035$ | $5384.02 \pm 0.3368$ | $6098.00 \pm 0.6047$ |
| M$^2$F-RNN-30day | $0.35 \pm 0.0044$ | $0.33 \pm 0.0037$ | $0.438 \pm 0.0029$ | $0.162 \pm 0.0080$ | $9885.76 \pm 0.4665$ | $12124.63 \pm 0.5682$ |
| **M$^2$F-PINN-30day** | $\mathbf{0.18 \pm 0.0031}$ | $\mathbf{0.16 \pm 0.0046}$ | $\mathbf{0.690 \pm 0.0018}$ | $\mathbf{0.618 \pm 0.0125}$ | $\mathbf{1.11 \pm 0.0224}$ | $\mathbf{1.50 \pm 0.0124}$ |
| M$^2$F-CNN-60day | $0.15 \pm 0.0023$ | $0.18 \pm 0.0042$ | $0.732 \pm 0.075$ | $0.570 \pm 0.0027$ | $5865.83 \pm 0.6012$ | $7121.22 \pm 0.6723$ |
| M$^2$F-RNN-60day | $0.38 \pm 0.0060$ | $0.34 \pm 0.0071$ | $0.120 \pm 0.0082$ | $0.044 \pm 0.0071$ | $9687.30 \pm 0.9111$ | $11918.33 \pm 0.5508$ |
| **M$^2$F-PINN-60day** | $\mathbf{0.19 \pm 0.0006}$ | $\mathbf{0.17 \pm 0.0007}$ | $\mathbf{0.752 \pm 0.0110}$ | $\mathbf{0.704 \pm 0.00758}$ | $\mathbf{2.25 \pm 0.0042}$ | $\mathbf{1.90 \pm 0.0103}$ |

ble. We additionally clarify that the 22-hour runtime reported for WenHai corresponds to CPU-only inference.

## A.8 ADDITIONAL MULTI-DAY EXPERIMENTAL RESULTS COMPARING DIFFERENT METHOD

Here, we provided the 7-day, 30-day, and 60-day performance results for M$^2$F-CNN, M$^2$F-RNN, and M$^2$F-PINN model in Table 6. All experiments were conducted with three independent runs to ensure statistical robustness. The results consistently show that M$^2$F-PINN outperforms the other two models across all evaluated forecasting horizons.

Figure 7 reports the layer-wise results of M$^2$F-PINN under 1-day, 7-day, 30-day, and 60-day forecasting settings. For M$^2$F-PINN-1day, we observe that as ocean depth increases, the RMSE of $U$ and $V$ gradually increases, while ACC decreases and PIC increases. Nevertheless, all metrics remain within a reasonable range, indicating stable performance. In contrast, for M$^2$F-PINN-7day, 30day, and 60day, the degradation of RMSE, ACC, and PIC with increasing depth becomes more pronounced and less stable compared to the 1-day model. This trend confirms that prediction becomes increasingly challenging at greater depths, especially for longer forecast horizons. Potential reasons include: (1) more complex physical processes governing deep-ocean circulation, (2) higher noise levels in deep-water observations, and (3) accumulated uncertainty in long-term autoregressive predictions.

## A.9 ADDTIONAL RESULTS

Figure 8 and 9 present visualizations of the $U$ and $V$ components across 13 vertical layers on September 6, 2006, respectively.

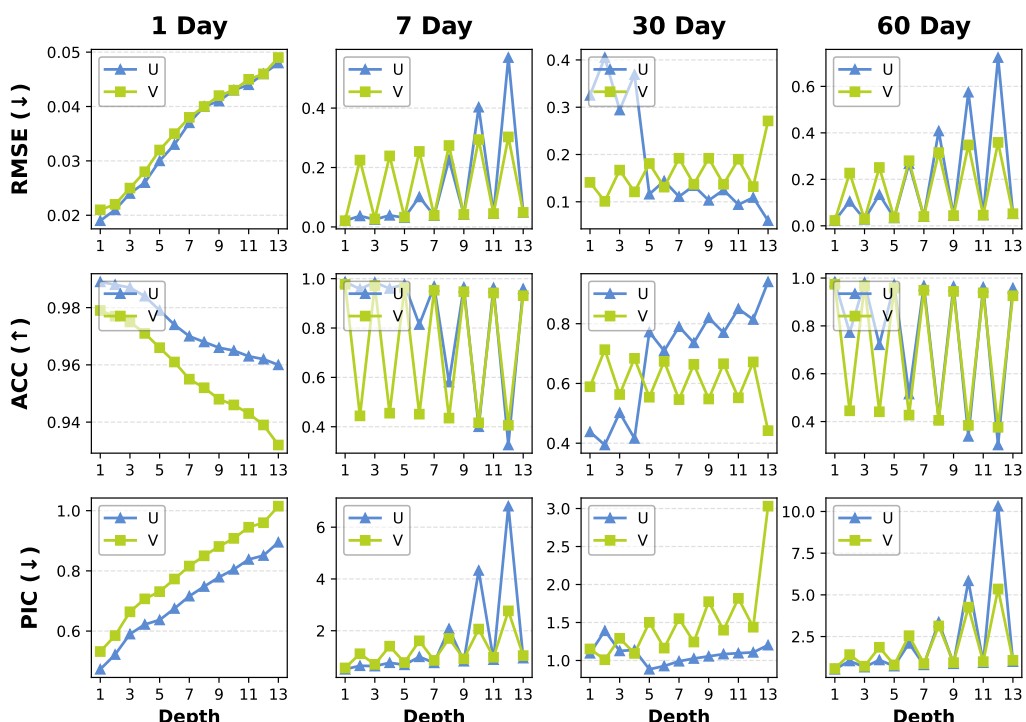

Figure 7: Comparison of Prediction Results for the M$^2$F-PINN-1DAY, 3-day, 7-day, and 60-day Models at 13 Ocean Depths.

## A.10 THE USE OF LARGE LANGUAGE MODELS (LLMs)

In this paper, Large Language Models (LLMs) were utilized to aid in polishing the English writing. Specifically, the LLM was employed to enhance the fluency, accuracy, and clarity of the English text.

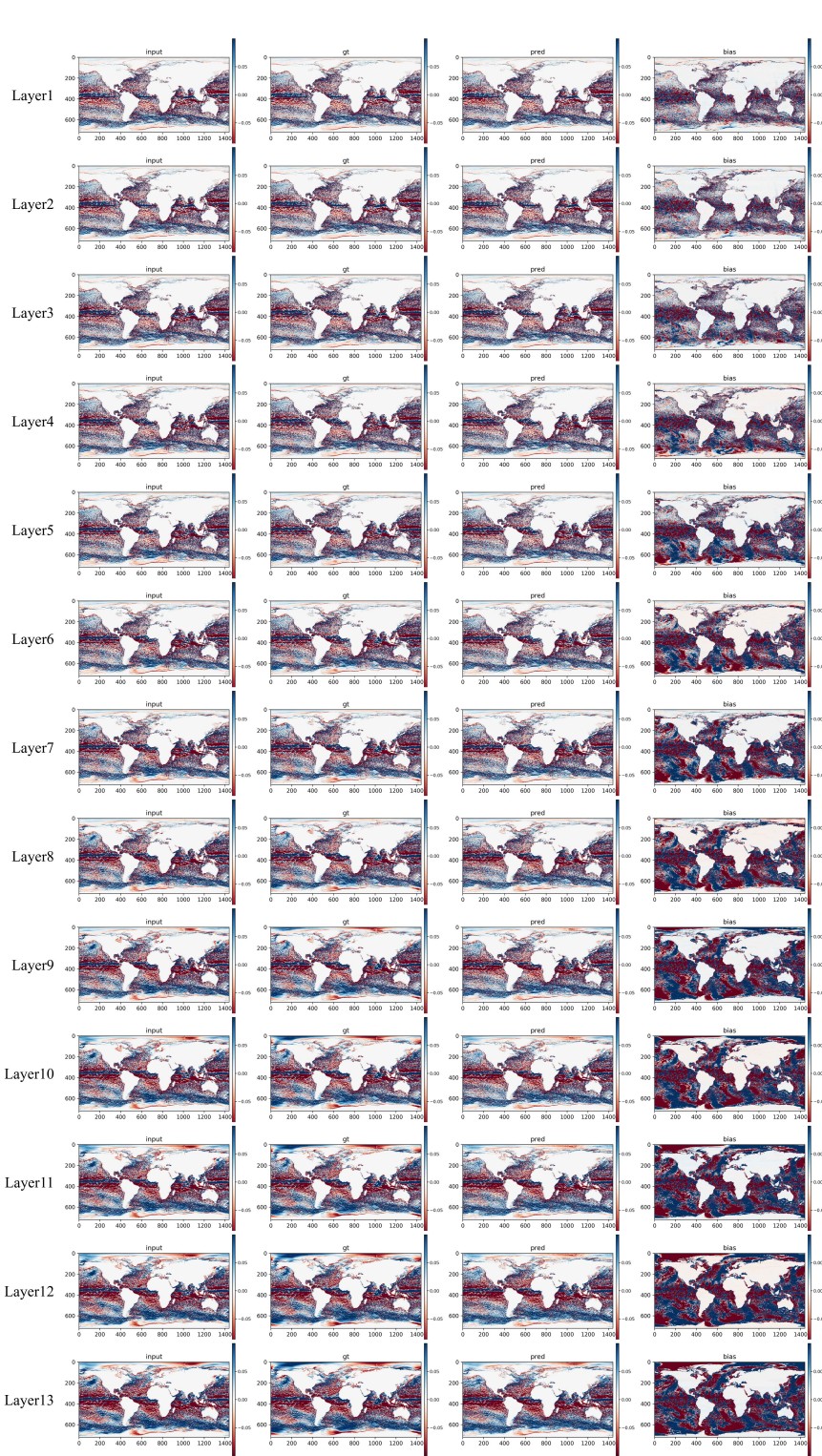

Figure 8: Visualization for $U$ component in 13 layers. In our vertical discretization scheme, layer1 to layer12 correspond to the 13 ocean layers from the surface downward, with layer0 representing the sea surface and layer12 indicating the deepest layer.

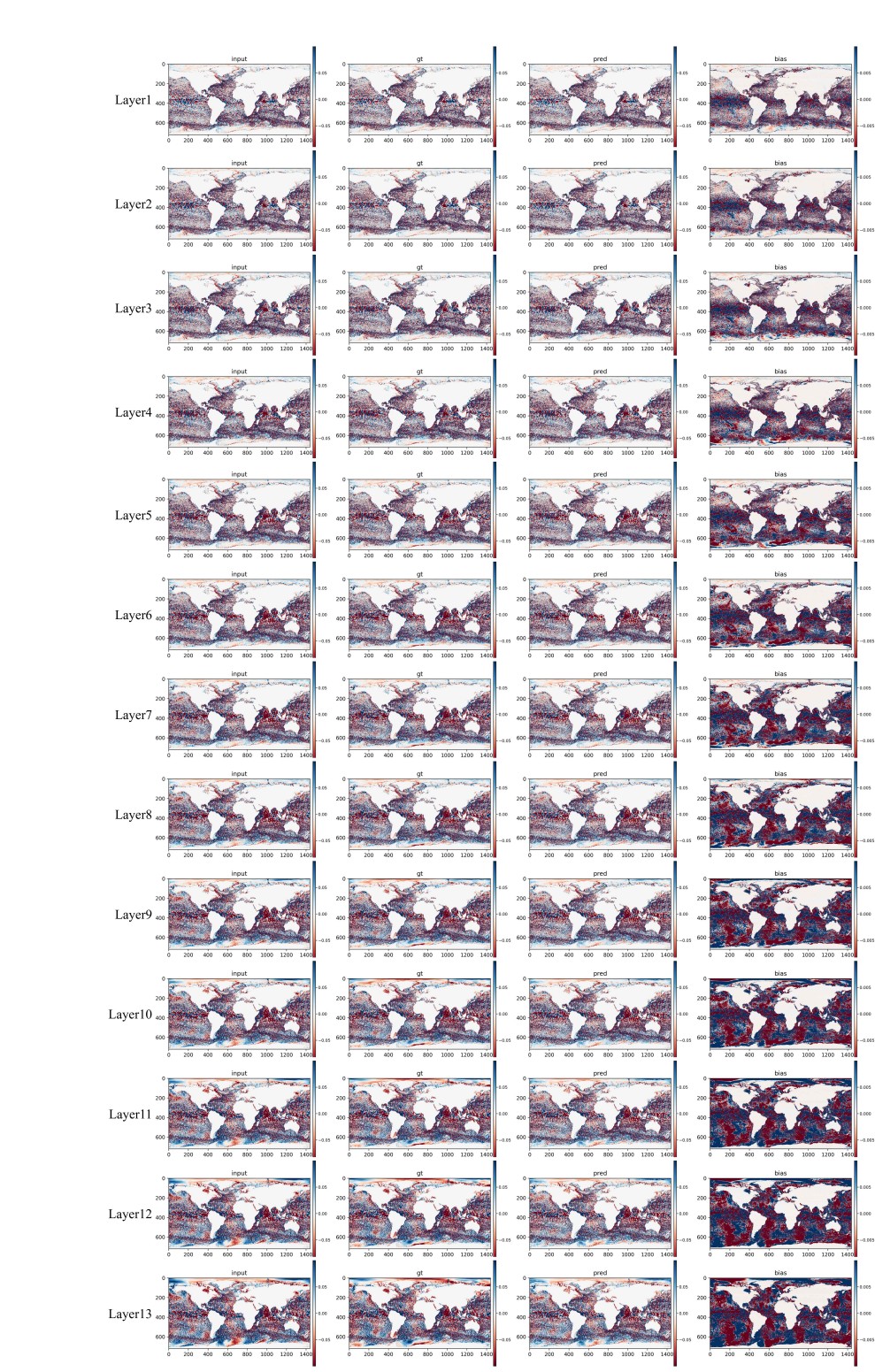

Figure 9: Visualization for $V$ component in 13 layers.

