# OpenReview forum: "M²F-PINN: A Multi-Scale Frequency-Domain Multi-Physics-Informed Neural Network for Ocean Forecasting"
_ICLR.cc/2026/Conference — Submitted to ICLR 2026_

### Official Review · Reviewer_ZXEn · 2025-10-26

**Soundness:** 2
**Presentation:** 3
**Contribution:** 3
**Rating:** 6
**Confidence:** 4

**Summary:**

This paper presents M²F-PINN, a Multi-Scale Frequency-domain Physics-Informed Neural Network designed for ocean current forecasting. The model integrates multi-frequency feature embeddings, momentum-equation-based physical constraints, and a 3D Swin Transformer backbone to capture multi-scale dynamics. Specifically, it introduces Fourier mapping to address spectral bias in the model, incorporates PDE residuals from zonal and meridional momentum equations as physics loss terms, and learns spatiotemporal evolution via a Transformer-based autoregressive framework. Experiments conducted on the GLORYS12 reanalysis dataset (2005–2008) show that the proposed model outperforms several baselines in both prediction accuracy and physical consistency.

**Strengths:**

(1) Multi-scale representation via frequency embeddings. The use of Gaussian Fourier features helps alleviate spectral bias and improves learning of high-frequency dynamics, a common limitation of PINNs and spatiotemporal neural operators.
(2) Across all evaluation metrics, M²F-PINN achieves higher accuracy and better stability compared to state-of-the-art models such as XiHe, and WenHai. The results suggest that the multi-scale frequency-domain learning strategy, combined with physics-informed constraints, effectively enhances both prediction accuracy.

**Weaknesses:**

(1) Experimental setup simplicity and comparison fairness. The experimental design focuses on a relatively simplified ocean forecasting setup. In contrast, baseline models such as XiHe and WenHai were originally developed for more complex, multi-variable, and higher-resolution scenarios. Using them under a simplified setting may reduce the fairness and persuasive power of the comparison, as these models are not optimized for such reduced configurations. However, the simplified experimental configuration in this paper makes the comparison less convincing.
(2) Limited novelty in methodological design. The integration of frequency-domain representations (Fourier features), NTK-inspired spectral components, and PDE-based physics losses is reasonable and well-motivated but not conceptually new. Similar combinations have appeared in recent physics-informed operator learning and PINNs literature. As a result, the innovation of M²F-PINN lies more in assembling existing techniques than in introducing a fundamentally new modeling principle or architecture.
(3) Minor typo. In line 48, the referenced model name “AI-GMOS” is a typo — it should be “AI-GOMS.”

**Questions:**

Since the GLORYS12 dataset is a reanalysis data rather than a pure numerical model output, it may not strictly obey the PDEs used for the physics-informed loss. Could the authors clarify whether the GLORYS fields are consistent with the momentum equations? If inconsistencies exist, how are they handled during optimization, and how does this affect model stability and interpretability?

---

> ### Author Response · Authors · 2025-11-23
> **Response to Reviewer ZXEn(Part 1)**
>
> ## **Global response**
>
> ---
>
> **Response to Summary and Strengths:** We appreciate your summary and positive assessment of our work. We are pleased that you recognize the strengths of our multi-scale frequency-domain physics-informed neural network model, particularly the multi-scale representation based on frequency embeddings and the model's advantages in terms of prediction accuracy and physical consistency. The author team has carefully discussed your comments and prepared targeted responses accordingly.
>
> ---
>
> ## **Weaknesses**
>
> ---
>
> **Response to Weakness-1:** We thank you for this insightful comment. As you correctly pointed out, XiHe and WenHai are highly influential and successful ocean forecasting models that provide accurate eddy-resolving predictions and have achieved excellent performance. In contrast to these systems, which primarily target eddy-related ocean phenomena, our work focuses on forecasting fundamental ocean state variables, in particular the horizontal velocity components ($U$ and $V$). As is well known, $U$ and $V$ are directly linked to a wide range of ocean processes such as acoustic propagation, eddy and meso-scale dynamics, and, more broadly, have important implications for climate and marine ecosystems. Therefore, accurate prediction of these basic dynamical variables is of fundamental importance, and addressing this need is the central objective of our study.
> Moreover, once reliable forecasts of the $U$ and $V$ fields are obtained, they can naturally serve downstream applications such as ocean acoustic field modeling and sound-channel axis prediction. Compared with the environmental configurations adopted in XiHe and WenHai, our experimental setup is explicitly constructed to emphasize the dominant factors driving variability in $U$/$V$, with a particular focus on rapidly evolving ocean currents at meso- and larger- scales. This setting imposes substantial challenges on $U$/$V$ prediction and thus provides a stringent testbed for our proposed approach.
>
> ---
>
> **Response to Weakness-2:** We sincerely thank you for this professional and important comment.
>
> The core contribution of this work lies in the targeted integration of deep learning advances into the geophysical domain, specifically ocean modeling. To the best of our knowledge, we are the first to combine an improved multi-scale Fourier feature mapping network with physics-informed neural networks (PINNs). This tailored integration substantially enhances both predictive accuracy and physical consistency for ocean surface velocity field forecasting. Through extensive experiments and ablation studies, we systematically demonstrate not only that the proposed approach is effective, but also why it is effective (e.g., it achieves better capture of multi-scale spectral energy transfer) and under which conditions it delivers the most significant gains (e.g., in rapidly evolving meso-scale and large-scale regimes). We believe this domain-specific, theoretically grounded fusion represents a meaningful advance at the intersection of scientific machine learning and Earth sciences.
>
> ---
>
> **Response to Weakness-3:** We thank you for the careful reading and professional comment.
>
> We sincerely apologize for the inadvertent typo in the manuscript. The correct name of our proposed model is **AI-GOMS** (not AI-GMOS). This error appeared in several places in the original submission and has now been corrected throughout the revised manuscript. Please see the updated PDF (e.g., line 128 in Section 3.1, line 256 in Section 4, and all figure captions) for the corrected acronym. We are grateful to the reviewer for catching this oversight.

---

> ### Author Response · Authors · 2025-11-23
> **Response to Reviewer ZXEn(Part 2)**
>
> ## **Questions**
>
> ---
>
> **Response to Question-1:** Thank you for this highly professional and insightful question.
>
> 1. GLORYS12 reanalysis is produced by the NEMO (Nucleus for European Modelling of the Ocean) dynamical model with 4D-Var data assimilation that incorporates a wide range of observations. As a result, its velocity fields are generally in strong agreement with the momentum equations. However, incremental adjustments introduced during the assimilation process can lead to minor violations of the continuous PDE constraints. In our experiments, the Physics-Informed Consistency (PIC) metric directly measures the residual of the momentum equations. For the GLORYS12 ($U$, $V$) fields used in this study, the mean PIC values for u and v are approximately 0.7–0.8 (in the normalized units defined in Section 4.2), which are several orders of magnitude lower than the PIC values typically observed for purely data-driven baselines (∼6000–7000). This confirms that the momentum equations used in our PINN are highly consistent with the reference GLORYS12 velocity fields.
> 2. To robustly handle any remaining minor inconsistencies, our training loss includes the standard PINN MSE physics residual term, which explicitly penalizes deviations from the momentum equations. Additionally, the high- and low-frequency Fourier feature mappings in our multi-scale architecture act as an effective spectral filter, naturally smoothing out high-frequency noise or assimilation-induced artifacts present in the GLORYS12 data.
> 3. While such small inconsistencies could in principle introduce noise into the training process, our multi-scale frequency-domain design significantly mitigates spectral bias, and the strong physics priors substantially enhance training stability. Embedding the governing laws—even when the reference data are not perfectly PDE-compliant—greatly improves generalization and interpretability, which is one of the key advantages of the physics-informed paradigm. We believe these design choices make our approach particularly well-suited for real-world reanalysis products such as GLORYS12.

---

### Official Review · Reviewer_8DXa · 2025-10-29

**Soundness:** 2
**Presentation:** 1
**Contribution:** 2
**Rating:** 2
**Confidence:** 3

**Summary:**

This paper introduces M2F-PINN, a multi-scale, frequency-domain physics-informed neural network for ocean forecasting. It combines Fourier representation learning to address spectral bias and uses 3D Swin Transformer to capture spatiotemporal patterns. M2F-PINN integrates physics-based priors through loss function about physical constraints. Experiments show it outperforms deep-learning baselines and ocean models.

**Strengths:**

1. The paper provides substantial theoretical support, explaining the impact of spectral bias on network training.
2. The ablation study examines the contribution of each major component (training B, training scale, training frequencies), which improves the transparency of the design.

**Weaknesses:**

1. The model introduces momentum equations to constrain the training of U and V, however, momentum equations are based on specific physical assumptions and simplifications (e.g., constant density). These equations primarily address large-scale oceanic flows, and cannot fully capture small-scale dynamics such as turbulence and eddies. Furthermore, the equation can only describe velocity variations accurately at higher resolutions. Utilizing this equation as a physical constraint may affect the precise modeling and prediction of small-scale flows.
2. The method utilizes Gaussian Fourier coordinate encoding to enhance the exposure of high-frequency information. However, this paper does not include experimental design to demonstrate how the method enables the model to capture ocean current features at different scales, nor does it modify the method based on the characteristics of ocean flow fields. It appears more aligned with application-oriented research in the geoscience rather than a fundamental advance in physical modeling.
3. Although the work includes several comparative experiments, there are still some issues to be addressed. a) There is a lack of comparison with recent baseline works, such as PINN and neural operator approaches from the past two years. b) The paper does not demonstrate how M2F-PINN compares with other methods in terms of performance at different prediction time horizons and depths. c) The ablation study does not clearly specify which modules the PINN-base and data-base ablations refer to.
4. The overall presentation of the paper is unclear, and the figures and tables are relatively rough. Some examples include: a) The phrase "learns frequency components multi-scales to improve multi-scale dynamics" in the abstract is grammatically incorrect. b) There is an issue with the citation format, leading to repeated author names. c) The borders of Tables 1 and 2 are unclear, and Table 3 exceeds the article width. d) The introduction of the NTK theory is abrupt, and its connection to the work is not explained. e) Only the momentum theorem is used, but why is the method called Multi-PINN? f) Many variables in the formulas are not explained, such as the variable t in line 218. g) The dataset description is unclear, such as the specific resolution (how many kilometers).

**Questions:**

Please refer to the weaknesses.

---

> ### Author Response · Authors · 2025-11-23
> **Response to Reviewer 8DXa(Part 1)**
>
> ## **Global response**
>
> ---
>
> **Response to Summary and Strengths:** We sincerely appreciate your thoughtful summary and positive evaluation of our work. We are pleased that you recognize the effectiveness of our proposed multi-scale frequency–domain and multi-physics–informed framework for ocean current forecasting, the supporting theory regarding spectral bias, and the comprehensive comparison with advanced ocean models such as XiHe and WenHai. Beyond the contributions outlined at the end of the Introduction, our work further advances the field through the following key components:
>
> 1. Multi-scale frequency–domain representation learning, which enables the model to capture ocean dynamics across a wide range of spatial scales.
> 2. Multiple PINN constraints that encode essential oceanic physical laws for flow dynamics, combined with an uncertainty-aware weighting strategy to adaptively balance the contributions of multiple loss terms.
> 3. A theoretical analysis grounded in Neural Tangent Kernel (NTK) theory, elucidating the impact of spectral bias on neural networks and motivating the use of multi-scale Fourier feature embeddings to mitigate this issue within PINNs.
>
> We are grateful for your recognition of these core aspects of the manuscript. Our author team has carefully considered your comments and prepared targeted responses accordingly.
>
> ---
>
> ## **Weaknesses:**
>
> ---
>
> **Response to Weaknesses-1:** Thank you for this professional comment. We agree with this observation. This study does not address small-scale flows (i.e., spatial resolutions below 1km; temporal resolutions from seconds to hours). Our work focuses on meso-scale (1-100km; days to months) and large-scale flows (over 100km; days to years). This scope is determined by our training data, which has a spatial resolution of 0.25° (approximately 13.88km to 27.75km) and a temporal resolution of one data point per day. Consequently, the equation adopted in this study is not intended for precise modeling or prediction of small-scale dynamics, but rather serves to constrain and enhance the representation of meso- and large-scale circulation features.
>
> ---
>
> **Response to Weaknesses-2:** Thank you for your professional comments. Following your valuable suggestion, we have added a comprehensive power spectrum analysis in the revised manuscript to strengthen the evaluation of multi-scale modeling capability. As shown in the updated Figure 6, which presents the multi-scale spectral comparison, our M$^2$F-PINN accurately captures both mesoscale (100–500 km) and large-scale (>500 km) oceanic variability. The predicted spectrum exhibits near-perfect overlap with the ground truth at these scales, demonstrating the model’s strong ability to recover multi-scale ocean dynamics. In contrast, the PINN variant without the frequency-domain module fails to capture these multi-scale structures, further highlighting the necessity of our proposed design. A dedicated subsection, “Power-Spectrum Analysis of M$^2$F-PINN,” has been added to the revised manuscript to detail this analysis.
>
> On the other hand, inspired by prior work, this study innovatively applies the encoding mechanism to the challenging new domain of complex ocean modeling and further introduces tailored improvements, whose effectiveness is validated through extensive experiments. Ocean circulation is inherently a multi-scale and multi-frequency dynamical system, and our proposed approach directly addresses this longstanding challenge in Earth system modeling. We believe that the application of physics-informed neural networks to such a complex geophysical setting provides a valuable paradigm and benchmark, which may inspire future developments in interdisciplinary research.

---

> ### Author Response · Authors · 2025-11-23
> **Response to Reviewer 8DXa(Part 2)**
>
> **Response to Weaknesses-3:**
>
> Thank you for these professional comments. Following your valuable suggestion, we address your comments as follows:
>
> a) First, regarding baselines, we will add comparisons with hard-constrain PINN (HardC) methods. We provide the updated Table 1 (shown below) in the paper. We clarify that Neural Operators were already included in our original comparison (Table 1), specifically FNO and MeshGraphNets.
>
> | Description     | $U$-RMSE(↓)     | $V$-RMSE(↓)     | $U$-ACC(↑)      | $V$-ACC(↑)       | $U$-PIC(↓)       | $V$-PIC(↓)        |
> | --------------- | --------------- | --------------- | --------------- | ---------------- | ---------------- | ----------------- |
> | CNN             | 5.36±0.1781     | 19.55±0.1917    | 0.942±0.0023    | 0.742±0.0026     | 7250.50±388.35   | 9802.44±272.30    |
> | M$^2$F-CNN      | 0.05±0.0079     | 0.06±0.0129     | 0.955±0.044     | 0.926±0.0053     | 4610.53±0.4166   | 5378.97±0.2349    |
> | RNN             | 10.16±8.3810    | 24.24±8.2873    | 0.862±0.1386    | 0.584±0.2757     | 7063.19±639.37   | 9562.37±449.80    |
> | M$^2$F-RNN      | 0.12±0.0144     | 0.12±0.0133     | 0.746±0.0015    | 0.595±0.0045     | 9686.31±0.0632   | 11917.8±0.4382    |
> | MeshGraphNets   | 0.04 ±  0.0047  | 0.04 ± 0.0060   | 0.938 ± 0.0031  | 0.920 ± 0.0025   | 8004.71 ± 0.6538 | 11558.25 ± 0.5353 |
> | FNO             | 0.14 ±  0.0049  | 0.11 ± 0.0052   | 0.035 ± 0.0045  | -0.030 ± 0.0025  | 7237.35 ± 1.0132 | 10686.15 ± 0.9500 |
> | **HardC**       | 0.067±0.0038    | 0.058±0.0024    | 0.661±0.0087    | 0.665±0.0073     | 1.40±0.0824      | 1.876±0.0730      |
> | XiHe            | 0.19            | 0.19            | 0.963           | 0.938            | 380907.58        | 441561.92         |
> | WenHai          | 0.17            | 0.16            | 0.903           | 0.923            | 1344.1           | 84768.29          |
> | **M$^2$F-PINN** | **0.03±0.0003** | **0.03±0.0001** | **0.972±0.008** | **0.953±0.0036** | **0.71±0.0032**  | **0.80±0.0063**   |
>
> b) Second, after careful analysis, we believe the 1-day results are representative of the model's performance. Due to computational resource limitations, we did not conduct experiments across all horizons for all models. However, to illustrate the trend, we provide the 1day, 7-day, 30-day, and 60-day results for M$^2$F-CNN, M$^2$F-RNN and M$^2$F-PINN model. All experiments were conducted with three independent runs to ensure statistical robustness. The results consistently show that M$^2$F-PINN outperforms the other two models across all evaluated forecasting horizons.
>
> | Description           | $U$-RMSE(↓)     | $V$-RMSE(↓)      | $U$-ACC(↑)       | $V$-ACC(↑)        | $U$-PIC(↓)      | $V$-PIC(↓)      |
> | --------------------- | --------------- | ---------------- | ---------------- | ----------------- | --------------- | --------------- |
> | M$^2$F-CNN-1day       | 0.05±0.0079     | 0.06±0.0129      | 0.955±0.044      | 0.926±0.0053      | 4610.53±0.4166  | 5378.97±0.2349  |
> | M$^2$F-RNN-1day       | 0.12±0.0144     | 0.12±0.0133      | 0.746±0.0015     | 0.595±0.0045      | 9686.31±0.0632  | 11917.8±0.4382  |
> | **M$^2$F-PINN-1day**  | **0.03±0.0003** | **0.03±0.0001**  | **0.972±0.008**  | **0.953±0.0036**  | **0.71±0.0032** | **0.80±0.0063** |
> | M$^2$F-CNN-7day       | 0.06±0.0057     | 0.06±0.062       | 0.928±0.0087     | 0.889±0.0041      | 4873.83±0.8112  | 5530.79±0.6901  |
> | M$^2$F-RNN-7day       | 0.34±0.0080     | 0.26±0.0059      | 0.470±0.0075     | 0.498±0.0064      | 9688.10±0.8991  | 16793.21±0.6815 |
> | **M$^2$F-PINN-7day**  | **0.13±0.0024** | **0.148±0.0053** | **0.841±0.0081** | **0.720±0.0051**  | **1.61±0.0082** | **1.27±0.0068** |
> | M$^2$F-CNN-30day      | 0.11±0.0120     | 0.13±0.0041      | 0.820±0.0077     | 0.741±0.0035      | 5384.02±0.3368  | 6098.00±0.6047  |
> | M$^2$F-RNN-30day      | 0.35±0.0044     | 0.33±0.0037      | 0.438±0.029      | 0.162±0.0080      | 9885.76±0.4665  | 12124.63±0.5682 |
> | **M$^2$F-PINN-30day** | **0.18±0.0031** | **0.16±0.0046**  | **0.690±0.0018** | **0.618±0.0125**  | **1.11±0.0224** | **1.50±0.0124** |
> | M$^2$F-CNN-60day      | 0.15±0.0023     | 0.18±0.0042      | 0.732±0.0075     | 0.570±0.0027      | 5865.83±0.6012  | 7121.22±0.6723  |
> | M$^2$F-RNN-60day      | 0.38±0.0060     | 0.34±0.0071      | 0.120±0.0082     | 0.044±0.0071      | 9687.30±0.9111  | 11918.33±0.5508 |
> | **M$^2$F-PINN-60day** | **0.19±0.0006** | **0.16±0.0007**  | **0.752±0.0110** | **0.704±0.00758** | **2.25±0.0042** | **1.90±0.0103** |
>
> The table below reports the depth-wise results for M$^2$F-PINN under the 1-day, 7-day, 30-day, and 60-day forecasting settings. We include the table here for clarity. In the revised manuscript, these results are presented in a more interpretable line-plot format in Appendix A.8.

---

> ### Author Response · Authors · 2025-11-23
> **Response to Reviewer 8DXa(Part 3)**
>
> For the 1-day horizon, the results show that as ocean depth increases, RMSE gradually increases, ACC decreases, and PIC increases. Nevertheless, all metrics remain within a reasonable range, indicating that M$^2$F-PINN maintains stable performance across depths. For the 7-day and 30-day horizons, the degradation in RMSE, ACC, and PIC becomes more pronounced and less stable compared with the 1-day case. This trend suggests that prediction becomes increasingly challenging at greater depths, especially for longer forecasting horizons. Potential reasons include more complex deep-ocean dynamical processes, higher noise levels in deep-water observations, and accumulated uncertainty in long-term autoregressive prediction.
>
> | M$^2$F-PINN-1day | mean  | depth1 | depth2 | depth3 | depth4 | depth5 | depth6 | depth7 | depth8 | depth9 | depth10 | depth11 | depth12 | depth13 |
> | ---------------- | ----- | ------ | ------ | ------ | ------ | ------ | ------ | ------ | ------ | ------ | ------- | ------- | ------- | ------- |
> | $U$-RMSE(↓)      | 0.035 | 0.019  | 0.021  | 0.024  | 0.026  | 0.030  | 0.033  | 0.037  | 0.040  | 0.041  | 0.043   | 0.044   | 0.046   | 0.048   |
> | $V$-RMSE(↓)      | 0.036 | 0.021  | 0.022  | 0.025  | 0.028  | 0.032  | 0.035  | 0.038  | 0.040  | 0.042  | 0.043   | 0.045   | 0.046   | 0.049   |
> | $U$-ACC(↑)       | 0.973 | 0.989  | 0.988  | 0.987  | 0.984  | 0.979  | 0.974  | 0.970  | 0.968  | 0.966  | 0.965   | 0.963   | 0.962   | 0.960   |
> | $V$-ACC(↑)       | 0.957 | 0.979  | 0.977  | 0.975  | 0.971  | 0.966  | 0.961  | 0.955  | 0.952  | 0.948  | 0.946   | 0.943   | 0.939   | 0.932   |
> | $U$-PIC(↓)       | 0.704 | 0.473  | 0.522  | 0.590  | 0.622  | 0.637  | 0.675  | 0.716  | 0.748  | 0.779  | 0.805   | 0.838   | 0.851   | 0.895   |
> | $V$-PIC(↓)       | 0.798 | 0.532  | 0.585  | 0.664  | 0.707  | 0.731  | 0.773  | 0.816  | 0.850  | 0.881  | 0.908   | 0.945   | 0.960   | 1.015   |
>
>
>
> | M$^2$F-PINN-7day | mean  | depth1 | depth2 | depth3 | depth4 | depth5 | depth6 | depth7 | depth8 | depth9 | depth10 | depth11 | depth12 | depth13 |
> | ---------------- | ----- | ------ | ------ | ------ | ------ | ------ | ------ | ------ | ------ | ------ | ------- | ------- | ------- | ------- |
> | $U$-RMSE(↓)      | 0.125 | 0.021  | 0.037  | 0.025  | 0.039  | 0.031  | 0.101  | 0.038  | 0.227  | 0.042  | 0.403   | 0.045   | 0.570   | 0.049   |
> | $V$-RMSE(↓)      | 0.142 | 0.021  | 0.225  | 0.027  | 0.239  | 0.033  | 0.254  | 0.039  | 0.274  | 0.042  | 0.294   | 0.045   | 0.303   | 0.049   |
> | $U$-ACC(↑)       | 0.834 | 0.987  | 0.958  | 0.985  | 0.960  | 0.977  | 0.814  | 0.969  | 0.581  | 0.965  | 0.400   | 0.963   | 0.326   | 0.959   |
> | $V$-ACC(↑)       | 0.715 | 0.977  | 0.444  | 0.972  | 0.455  | 0.963  | 0.451  | 0.953  | 0.435  | 0.947  | 0.416   | 0.942   | 0.406   | 0.931   |
> | $U$-PIC(↓)       | 1.604 | 0.510  | 0.647  | 0.633  | 0.762  | 0.683  | 0.992  | 0.759  | 2.077  | 0.822  | 4.329   | 0.883   | 6.812   | 0.941   |
> | $V$-PIC(↓)       | 1.267 | 0.563  | 1.110  | 0.700  | 1.403  | 0.771  | 1.610  | 0.853  | 1.702  | 0.915  | 2.064   | 0.975   | 2.767   | 1.039   |
>
>
>
> | M$^2$F-PINN-30day | mean  | depth1 | depth2 | depth3 | depth4 | depth5 | depth6 | depth7 | depth8 | depth9 | depth10 | depth11 | depth12 | depth13 |
> | ----------------- | ----- | ------ | ------ | ------ | ------ | ------ | ------ | ------ | ------ | ------ | ------- | ------- | ------- | ------- |
> | $U$-RMSE(↓)       | 0.184 | 0.325  | 0.405  | 0.294  | 0.369  | 0.116  | 0.144  | 0.111  | 0.134  | 0.103  | 0.125   | 0.094   | 0.109   | 0.060   |
> | $V$-RMSE(↓)       | 0.161 | 0.141  | 0.101  | 0.167  | 0.121  | 0.181  | 0.131  | 0.192  | 0.137  | 0.192  | 0.137   | 0.190   | 0.132   | 0.271   |
> | $U$-ACC(↑)        | 0.688 | 0.438  | 0.394  | 0.502  | 0.416  | 0.772  | 0.709  | 0.790  | 0.736  | 0.820  | 0.770   | 0.850   | 0.814   | 0.940   |
> | $V$-ACC(↑)        | 0.605 | 0.589  | 0.713  | 0.563  | 0.684  | 0.554  | 0.674  | 0.546  | 0.664  | 0.548  | 0.666   | 0.552   | 0.672   | 0.442   |
> | $U$-PIC(↓)        | 1.086 | 1.095  | 1.391  | 1.126  | 1.138  | 0.884  | 0.927  | 0.990  | 1.026  | 1.055  | 1.084   | 1.096   | 1.107   | 1.202   |
> | $V$-PIC(↓)        | 1.497 | 1.152  | 1.010  | 1.292  | 1.093  | 1.502  | 1.163  | 1.551  | 1.243  | 1.774  | 1.399   | 1.818   | 1.438   | 3.031   |

---

> ### Author Response · Authors · 2025-11-23
> **Response to Reviewer 8DXa(Part 4)**
>
> | M$^2$F-PINN-60day | mean  | depth1 | depth2 | depth3 | depth4 | depth5 | depth6 | depth7 | depth8 | depth9 | depth10 | depth11 | depth12 | depth13 |
> | ----------------- | ----- | ------ | ------ | ------ | ------ | ------ | ------ | ------ | ------ | ------ | ------- | ------- | ------- | ------- |
> | $U$-RMSE(↓)       | 0.191 | 0.022  | 0.105  | 0.028  | 0.134  | 0.034  | 0.267  | 0.040  | 0.408  | 0.044  | 0.575   | 0.047   | 0.724   | 0.052   |
> | $V$-RMSE(↓)       | 0.158 | 0.023  | 0.227  | 0.029  | 0.251  | 0.035  | 0.280  | 0.040  | 0.315  | 0.044  | 0.348   | 0.046   | 0.359   | 0.052   |
> | $U$-ACC(↑)        | 0.758 | 0.984  | 0.772  | 0.981  | 0.721  | 0.973  | 0.515  | 0.968  | 0.414  | 0.965  | 0.338   | 0.962   | 0.302   | 0.958   |
> | $V$-ACC(↑)        | 0.703 | 0.974  | 0.445  | 0.967  | 0.441  | 0.958  | 0.427  | 0.949  | 0.405  | 0.944  | 0.384   | 0.938   | 0.376   | 0.926   |
> | $U$-PIC(↓)        | 2.252 | 0.536  | 1.035  | 0.666  | 1.095  | 0.726  | 2.090  | 0.802  | 3.367  | 0.868  | 5.843   | 0.931   | 10.319  | 0.995   |
> | $V$-PIC(↓)        | 1.889 | 0.582  | 1.415  | 0.722  | 1.857  | 0.799  | 2.551  | 0.876  | 3.134  | 0.937  | 4.263   | 0.995   | 5.353   | 1.076   |
>
> c) Finally, we apologize for the oversight in the original manuscript, where we did not clearly describe the ablated modules. In the revised version, we have provided a detailed and explicit explanation of this component of our study. We clarify: PINN-Base ablates the Fourier Embedding module. Data-Base ablates both the PINN module and the Fourier Embedding module. See line 352 of the revised manuscript for details.
>
> ---
>
> **Response to Weaknesses-4:** Thank you for these professional comments.
>
> a) We revised the "learns frequency components multi-scales to improve multi-scale dynamics" to "learns multi-scale frequency components to enhance the modeling of multi-scale dynamics."
>
> b) Regarding the citations: Repetition of author names occurs because these works originate from the same research groups. [Chen et al., 2023b, 2024] are from the **FuXi team**. [Wang et al., 2021, 2022] are from the same **PINN research group** (Wang and Perdikaris). The works [Zhong et al., 2024a, 2024b, 2024c] are all part of the **FuXi series**.
>
> c) Thank you for this. We will correct the typos in Tables 1, 2, and 3 in the revised manuscript.
>
> d) We used the NTK-inspired spectral analysis to explain how Fourier embedding reshapes the learning rates, enabling stable network convergence.
>
> e) Thank you for your helpful remark. In our work, we formulate momentum constraints in two spatial directions, and therefore adopt the prefix multi- to indicate that the framework incorporates more than one physical momentum equation.
>
> f) We apologize for the oversight. We will add the following definitions: In Line 196, $t$ is the time in error dynamics and $e(t)$ is the error vector. In Line 103, $Y$ is the training target. In Line 207, $w_k$ are the frequency modes. In Eq. 14, $\Lambda$ (eigenvalue matrix), $Q$ (eigenvector matrix), $q_i$ (eigenvector), and $\lambda_i$ (eigenvalue).
>
> g) We thank the reviewer. The dataset resolution is 0.25°, which is 27.75km (latitude) and 13.88km to 27.75km (longitude).

---

### Official Review · Reviewer_3uEE · 2025-10-30

**Soundness:** 2
**Presentation:** 2
**Contribution:** 2
**Rating:** 4
**Confidence:** 3

**Summary:**

A novel model for global ocean current forecasting that aims to address the limitations of conventional Physics-Informed Neural Networks (PINNs) in capturing multi-scale variability and mitigating spectral bias

**Strengths:**

1.Achieves significantly lower Physical Inconsistency Coefficient (PIC) values, confirming that the predictions adhere more closely to the physical laws compared to all data-driven methods.

2.Demonstrates state-of-the-art performance, achieving the lowest RMSE and highest ACC across short- and long-range forecasts (up to 60 days), outperforming both deep learning baselines and competitive ocean models (e.g., XiHe and WenHai)

**Weaknesses:**

1.The framework is a novel combination, but the individual components—the 3D Swin Transformer and the Fourier Feature Embedding—are adapted from existing, established methods in computer vision and PINN literature, respectively.

2.The PDEs used for the PINN loss omit crucial terms (e.g., Coriolis force, pressure gradients) fundamental to ocean dynamics, potentially limiting the physical fidelity compared to full numerical models.

3.The paper lacks a comparison of training and inference speed against state-of-the-art models, which is essential for assessing the practical viability of the complex 3D Swin Transformer architecture.

**Questions:**

1.The PINN approach enforces constraints from the governing physical equations ($L_{PDE}$). However, real-world data (GLORYS12 reanalysis) inherently contains noise and discrepancies, and, the simplified PDEs used in $M^{2}F$-PINN are themselves an approximation of the full, complex ocean dynamics (the "reality gap"). Does enforcing an inexact or simplified physical constraint ultimately suppress the model's ability to learn real-world phenomena present in the data but not captured by the $\mathcal{L}_{PDE}$?

2. How to Ensure Inaccurate PINN Constraints Provide Model Gain, and Under What Conditions?

3.Given that the ablation study shows a marginal performance difference between $M^{2}F$-PINN and variants with non-trainable Fourier parameters, could the authors provide a stronger justification (e.g., analysis of learned B-matrix structures, convergence rates, or visualization of learned spectral coverage) for making these parameters adaptable and adding implementation complexity?

---

> ### Author Response · Authors · 2025-11-23
> **Response to Reviewer 3uEE(Part 1)**
>
> ## **Global response**
>
> ---
>
> **Response to Summary:** Thank you for your insightful summary of our work. In the context of short- and medium-term ocean current forecasting, our objective is to develop a more accurate prediction framework by (i) mitigating spectral bias through multi-scale frequency-domain feature learning, and (ii) enforcing physical consistency via a PINN formulation that embeds two key momentum equations. Together, these components enable the model to capture essential multi-scale ocean dynamics while maintaining physical constraints.
>
> We sincerely appreciate your recognition of the potential of multi-scale frequency-based modeling for global ocean current prediction. This encouragement motivates us to further explore and advance this research direction in future work.
>
> ---
>
> ## **Weaknesses**
>
> ---
>
> **Response to Weaknesses-1:** Thank you for your constructive and professional feedback. Motivated by the need for accurate global ocean current prediction, we seek feasible solutions from the computational modeling and machine learning community. Inspired by the long-range temporal modeling capability of the 3D Swin Transformer, we design a neural network to effectively exploit the historical temporal patterns of the $U$/$V$ variables, and therefore adopt a Transformer-based architecture for accurate prediction of ocean flow fields at different depth levels. In practice, we observe that land grid points introduce spurious signals that interfere with representation learning, so we incorporate a land–sea mask module. Furthermore, by analyzing the evolution characteristics of the $U$/$V$ current fields, we find that they exhibit fast and complex variations with prominent high-frequency components. This motivates us to employ Fourier-based feature extraction and to explore a multi-scale adaptive Fourier representation to mitigate spectral bias and better capture the multi-scale structure of global ocean current data.
>
> ---
>
> **Response to Weaknesses-2:** Thank you for your thoughtful and professional comments. As you correctly pointed out, the Coriolis force and pressure gradients are crucial for ocean dynamics. Here, we clarify why they are omitted in our current formulation. On the one hand, in this study we focus on an autoregressive forecasting model for only two variables, $U$ and $V$, and neither the Coriolis term nor the pressure gradient term is explicitly provided as input to the network. On the other hand, the simplified governing equation we adopt is empirically more consistent with the GLORYS12 reanalysis dataset used for training and evaluation. The experimental results show that our model already achieves competitive accuracy, and the PIC metric further confirms its physical fidelity.
>
> That said, the work you mentioned provides important inspiration for us. In future research targeting the prediction of additional dynamical variables and more complex parameterizations, we plan to explore incorporating the Coriolis force and pressure gradient terms in a principled manner.
>
> ---
>
> **Response to Weaknesses-3:** Thank you for your insightful comment. We apologize for not providing this information earlier. We have now added a comprehensive comparison of training and inference efficiency between our proposed M$^2$F-PINN and the state-of-the-art baselines. Since neither the XiHe nor WenHai papers report training time and only pre-trained weights are publicly available, we were able to evaluate these models only in inference mode. Consequently, we mark their training time as 'N/A' in our comparison table. For completeness, all runtime comparisons are summarized in Table 5 of the Appendix A.7. We clarify that the 22-hour runtime reported for WenHai corresponds to CPU-only inference. The updated manuscript now includes this table to facilitate a transparent and fair comparison of computational efficiency across all methods in Appendix A.7.
>
> | Description   | Training Time(Hours) | Test Time(Hours) |
> | ------------- | -------------------- | ---------------- |
> | CNN           | 11                   | 0.5              |
> | M$^2$F-CNN    | 12                   | 0.5              |
> | RNN           | 23                   | 0.52             |
> | M$^2$F-RNN    | 23                   | 0.52             |
> | MeshGraphNets | 0.25                 | 1.0              |
> | FNO           | 0.33                 | 0.27             |
> | HardC         | 0.8                  | 0.77             |
> | XiHe          | N/A                  | 3.5              |
> | WenHai        | N/A                  | 22$^1$           |
> | M$^2$F-PINN   | 21                   | 0.52             |

---

> ### Author Response · Authors · 2025-11-23
> **Response to Reviewer 3uEE(Part 2)**
>
> ## **Questions**
>
> ---
>
> **Response to Questions-1:** Thank you for your professional comments and careful consideration. Your insights are highly valuable for improving our manuscript. After thorough analysis and extensive comparative experiments, we conclude that imposing simplified physical constraints does not suppress the model’s ability to learn data-driven representations of real-world phenomena. Specifically, as shown in Table 3, we include a purely data-driven variant (“Data-Base”), whose RMSE, ACC, and PIC performance is consistently worse than that of the other two variants reported in the same table. This empirically supports our claim. For completeness, Table 3 of the manuscript is provided below.
>
> | Description     | $U$-RMSE(↓) | $V$-RMSE(↓) | $U$-ACC(↑) | $V$-ACC(↑) | $U$-PIC(↓) | $V$-PIC(↓) |
> | --------------- | ----------- | ----------- | ---------- | ---------- | ---------- | ---------- |
> | **M$^2$F-PINN** | **0.03**    | **0.03**    | **0.972**  | **0.953**  | **0.71**   | **0.80**   |
> | PINN-Base       | 0.04        | 0.04        | 0.965      | 0.941      | 86.94      | 57.79      |
> | Data-Base       | 0.05        | 0.05        | 0.951      | 0.914      | 6127.38    | 7434.28    |
>
> ---
>
> **Response to Questions-2:** Thank you for your professional and insightful comments. Our goal is not to force the model to strictly obey a potentially inaccurate simplified PDE. Instead, we use the PDE to encode the fundamental physical principles (e.g., momentum conservations) that act as inductive biases to guide the learning process. Even if simplified, such PDEs provide a strong form of structured regularization that improves physical plausibility and stabilizes extrapolation. In our framework, an imperfect physical constraint can still offer significant benefits, provided several key conditions are met:
>
> 1. **Scale consistency with the dominant dynamics.**
>    The simplified momentum equations used in our model primarily describe large- and meso-scale quasi-geostrophic ocean circulation. Our 0.25°-resolution GLORYS12 dataset is designed to capture precisely these scales, ensuring that the physics term aligns with the dominant flow regimes represented in the data.
> 2. **Balanced weighting between data loss and physics loss.**
>    We assign the simplified physics loss a *moderate* weight so that it serves as a guiding prior rather than an overly strict constraint. This allows the model to optimally balance fundamental physical principles with empirical patterns present in the observational data.
> 3. **Physics as soft constraints rather than exact equations.**
>    We do not expect the network to exactly satisfy the simplified PDE. Instead, the physics term continuously nudges the learning trajectory toward regions of the solution space that are physically reasonable.
>
> As a result, the final model learns a representation that inherits the large-scale dynamics described by the simplified PDE, while the data-driven component corrects its deficiencies. This leads to a more consistent and accurate dynamical model.
>
> ---
>
> **Response to Questions-3:** We appreciate your insightful observation. As you correctly pointed out, the performance gaps between M$^{2}$F-PINN and the variants with non-trainable Fourier parameters are indeed small. This prompted us to re-evaluate the relevance of the ablations “w/o training B,” “w/o training scale,” and “w/o training frequencies.” We agree that these components contribute only marginally to performance improvements and, more importantly, are not central to our main contributions—namely, the multi-scale frequency-domain learning and the multi-PINN-constrained forecasting framework. Following your valuable suggestion, we have moved these three less central ablation studies from Table 3 to the Appendix A.5. This adjustment preserves the detailed results while improving the rigor, clarity, and focus of the main paper, helping readers understand the core innovations.

---

### Official Review · Reviewer_JjqD · 2025-11-02

**Soundness:** 2
**Presentation:** 2
**Contribution:** 2
**Rating:** 2
**Confidence:** 3

**Summary:**

This paper presents M2F-PINN, a Multi-scale Frequency-domain Multi-Physics-Informed Neural Network for large-scale ocean current forecasting. The authors aim to address two known issues in ocean forecasting models: (1) deep models’ difficulty in learning both low- and high-frequency ocean dynamics due to spectral bias, and (2) the lack of physical consistency in purely data-driven ocean models. M2F-PINN propose several crucial components: multi-scale Fourier feature embeddings, a 3D Swin Transformer backbone for spatiotemporal feature extraction, and multiple Physics-Informed Neural Network (PINN) modules. The model is trained on the GLORYS12 reanalysis dataset (global ocean, 2005–2008) and evaluated against CNNs, RNNs, MeshGraphNets, Fourier Neural Operators, and state-of-the-art ocean models. M2F-PINN achieves the best results across 1–60-day prediction horizons, outperforming strong baselines and showing improved long-term physical consistency.

**Strengths:**

- The paper systematically combines frequency-domain embeddings, physics-informed losses, and Transformer architectures in a coherent way. Although each component exists independently, their integration into an end-to-end framework is well-executed.

- The NTK-based spectral analysis provides a clear explanation of how Fourier embeddings reshape learning rates across frequency bands, offering some theoretical insight beyond empirical results.

**Weaknesses:**

- The proposed components—multi-scale Fourier features, PINN-based physical losses, and uncertainty-weighted multi-task optimization—are all mature and widely used in physics-informed and climate modeling literature (e.g., FNO, ClimODE, LangYa, frequency-domain PINNs). The contribution is largely an engineering combination rather than a fundamentally new algorithmic advance. The claimed innovation in “frequency-domain multi-PINN” lacks distinctive technical depth beyond prior works.

- The physics component only implements two simplified momentum equations for (U, V), omitting key coupled variables such as temperature, salinity, and density that govern realistic ocean circulation. As a result, the “multi-physics” claim is overstated—the scope is narrow and its real-world interpretability limited.

- CNN and RNN serve as trivial references and do not represent current forecasting standards. Comparisons to XiHe and WenHai are insufficiently documented—there is no clarification whether they were re-trained under the same data regime or results are copied from publications. Missing comparisons with more recent AI-based earth system models (e.g., GraphCast, Fuxi, Pangu) weaken the empirical credibility.

- Despite the title emphasizing forecasting, the paper fails to specify how prediction is carried out.The architecture uses a Swin Transformer encoder-decoder, but it is unclear whether forecasts are autoregressive (iteratively rolling forward) or direct multi-step predictions.The “Algorithm 1” description is ambiguous: while labeled “autoregressive,” it lacks any explicit temporal unrolling or teacher-forcing details. Without a precise description of forecast horizon handling, it is hard to interpret the reported 1-, 7-, and 30-day results or to assess generalization stability.

- The paper asserts that Fourier embeddings capture “multi-scale oceanic structures,” yet provides no frequency-space visualizations or power-spectrum analysis to substantiate this claim. Likewise, the physical residuals (PIC) are introduced, but no concrete examples of physically consistent predictions (e.g., energy or vorticity preservation) are shown.

**Questions:**

- How exactly are forecasts generated? Is the model trained in an autoregressive fashion (iterative multi-step rollout) or direct multi-horizon regression? If autoregressive, how is error accumulation handled?

- Given that only U and V momentum equations are included, how can the model capture coupled dynamics driven by temperature and salinity gradients? Would incorporating the full Navier–Stokes or continuity equations improve realism?

---

> ### Author Response · Authors · 2025-11-23
> **Response to Reviewer JjqD(Part 1)**
>
> ## **Global response**
>
> ---
>
> **Response to Summary and Strengths: **Thank you for your thoughtful summary and positive assessment of our work. We are pleased that you recognize the value of our frequency-domain embedding and multi-physics-informed Transformer framework for spatiotemporal ocean forecasting. Our goal is to develop a multi-scale frequency-learning and multi-PINN architecture that effectively mitigates two central challenges in ocean prediction: spectral bias and insufficient physical consistency. Beyond the contributions summarized at the end of the Introduction, our work introduces several key innovations:
>
>  (1) a multi-scale frequency-domain representation that enables the model to capture rich cross-scale ocean dynamics;
>  (2) multiple PINN constraints that enforce ocean physical laws on the predicted flow fields, together with an uncertainty-aware weighting mechanism that adaptively balances the various losses; and
>  (3) a theoretical motivation grounded in Neural Tangent Kernel (NTK) analysis, which clarifies how spectral bias affects neural network training and why multi-scale Fourier embeddings help alleviate this issue within PINNs.
>
> Our author team carefully discussed your comments and has provided detailed, targeted responses below.
>
> ---
>
> ## **Weaknesses**
>
> ---
>
> **Response to Weaknesses-1:** Thank you for this detailed and thoughtful comment. We would like to clarify our technical contributions in more detail. In this work, we exploit multi-scale frequency-domain representations to extract rich information from ocean current fields. To the best of our knowledge, we are the first to combine an improved multi-scale Fourier feature mapping network with a physics-informed neural network (PINN) to address the challenging problem of accurately predicting key variables in ocean circulation.
>
> Importantly, this is not a simple architectural concatenation, but a carefully designed integration tailored to high-precision forecasting. On the multi-scale Fourier side, we introduce two adaptively learned low- and high-frequency components, which simultaneously alleviate spectral bias and enable the model to capture cross-scale interactions in ocean dynamics. On the PINN side, we learn the full set of physical coefficients associated with the two momentum equations over the entire spatiotemporal domain. In addition, we introduce several engineering-level innovations: for the 3D Swin Transformer backbone, we incorporate a land–sea mask to reduce computational cost while enhancing the internal coherence of ocean-only regions.
>
> ---
>
> **Response to Weaknesses-2: **Thank you for your insightful comments. Our choice of applying PINN-based constraints only to the two momentum equations is deliberate and technically motivated. Please allow us to further clarify the rationale behind this design. As you correctly pointed out, temperature, salinity, and density have an important influence on ocean circulation. However, the scalar fields (temperature, salinity, and density) exhibit relatively smooth spatial variations compared with the velocity components. Empirically, standard data-driven backbone already captures these low-frequency patterns effectively without requiring additional PDE supervision. In contrast, the horizontal velocity fields contain sharper gradients and more pronounced high-frequency dynamics that benefit substantially from explicit physical regularization. Therefore, to streamline the model architecture and improve computational efficiency, we enforce PINN-based constraints only on the $U$/$V$ momentum variables, whose spatial fields contain richer high-frequency information and thus allow the PINN term to encode more additional, complementary dynamics.
>
> In addition, our “multi-physics” formulation refers to integrating learned physical dynamics (momentum equations) with the prediction of the $U$- and $V$-velocity components within a unified framework. The prefix *multi-* denotes ‘more than one’.

---

> > ### Author Response · Authors · 2025-11-23
> > **Response to Reviewer JjqD(Part 2)**
> >
> > **Response to Weaknesses-3:** Thank you for your professional review and valuable suggestions, which are very helpful for improving our manuscript.
> >
> > First, please allow us to clarify the motivation for choosing CNN and RNN as baselines. CNNs and RNNs are two fundamental neural architectures, and in our setting they serve as reference baselines for the performance that can be achieved by standard data-driven predictors. Comparisons “above” these baselines allow us to clearly isolate and quantify the contribution of our algorithmic improvements and specialized architectural components.
> >
> > Second, following your suggestion, we have added a comparison with XiHe and WenHai. Both are large-scale ocean models with publicly available pretrained weights. In our experiments, we directly use their released models for inference on our dataset and report the resulting performance, ensuring a fair and realistic evaluation of these off-the-shelf ocean foundation models.
> >
> > Finally, we would like to explain why GraphCast, Fuxi, and Pangu are not included in our comparison. These three models are highly influential atmospheric forecasting systems and have demonstrated impressive performance, primarily targeting variables such as wind speed and surface pressure. However, they are not designed to predict oceanic flow fields (e.g., horizontal currents), which is the focus of our study. Due to this mismatch in prediction targets, a direct quantitative comparison would be inappropriate and potentially misleading. Nevertheless, the ideas and successes of these works have been an important source of inspiration for our research.
> >
> > Once again, we sincerely appreciate your constructive comments. We have further enriched and refined our comparative experiments accordingly, which we believe makes the revised manuscript more rigorous and comprehensive.
> >
> > ---
> >
> > **Response to Weaknesses-4:** Thank you for your valuable comment. We apologize for the oversight in the manuscript and would like to provide a more detailed clarification below. Our multi-day forecasting is implemented through direct multi-step regression, not iterative roll-out. Specifically, we independently train models for 1-day, 7-day, 30-day, and 60-day lead times, each optimized to predict its target horizon directly. Thank you for pointing out this issue and the lack of clarity. Following your valuable suggestions, we have revised the manuscript to clarify this setting and explicitly add the explanation on Page 7, Line 356 in the revised manuscript. We believe that, with your guidance, the rigor of the manuscript will be further improved.
> >
> > ---
> >
> > **Response to Weaknesses-5:** Thank you for your professional and insightful comments. Following your valuable suggestion, we have added a comprehensive power spectrum analysis in the revised manuscript to strengthen the evaluation of multi-scale modeling capability. As shown in the updated Figure 6, which presents the multi-scale spectral comparison, our M$^2$F-PINN accurately captures both mesoscale (100–500 km) and large-scale (>500 km) oceanic variability. The predicted spectrum exhibits near-perfect overlap with the ground truth at these scales, demonstrating the model’s strong ability to recover multi-scale ocean dynamics. In contrast, the PINN variant without the frequency-domain module fails to capture these multi-scale structures, further highlighting the necessity of our proposed design. A dedicated subsection, “Power-Spectrum Analysis of M$^2$F-PINN,” has been added to the revised manuscript to detail this analysis.
> >
> > Regarding physical consistency, we follow the definition in the manuscript: $\text{PIC}(v,t) = \frac{1}{n}\sum_{i=1}^{n} (f(v_{target})-f(v_{pred}))^{2}$. Since this operator contains second-order spatial derivatives of the horizontal velocities, PIC naturally reflects discrepancies in the predicted dynamical energy and thus serves as an indicator of physical consistency. Therefore, to some extent, the PIC value serves as a proxy for the energy prediction bias. In our manuscript, we quantify physical consistency through the magnitude of PIC values reported in Tables 1–3 and the corresponding heatmaps in Figures 4, 5, 8, and 9.
> >
> > Once again, we sincerely appreciate your professional and insightful suggestions, which are highly valuable for enriching and improving the manuscript. We believe that incorporating the additional comparative results you recommended will further substantiate our claims and enhance the overall rigor of the manuscript.

---

> ### Author Response · Authors · 2025-11-23
> **Response to Reviewer JjqD(Part 3)**
>
> ## **Questions**
>
> ---
>
> **Response to Questions-1:** We apologize for the oversight and lack of rigor in the manuscript. Below, we provide a detailed explanation of how the prediction results are generated. Please refer to Response to Weaknesses-4 for a detailed clarification of this point. Specifically, our multi-day forecasting models are trained using direct multi-day regression rather than iterative rollout. In other words, we independently train separate models for 1-day, 7-day, 30-day, and 60-day prediction horizons, instead of recursively applying a 1-day model to obtain longer-term forecasts.
>
> ---
>
> **Response to Questions-2:** Thank you for your thoughtful and professional comments. Our study focuses specifically on the temporal forecasting of the horizontal velocity components $U$ and $V$. Therefore, the model is not intended to capture the coupled thermodynamic dynamics driven by temperature and salinity gradients, as temperature and salinity prediction is outside the scope of this work. We thank you for directing our attention to the possibility of incorporating the full PDE system. After careful consideration, we believe that introducing the complete Navier–Stokes equations or the continuity equation is unlikely to further improve the physical fidelity of our model, primarily for the following two reasons:
>
> (1) Data source limitations. The training data are derived from the GLORYS12 reanalysis product, which is not a purely numerical simulation output. Reanalysis fields assimilate observations and therefore cannot strictly satisfy the full Navier–Stokes or continuity equations in their original form.
>
> (2) Input variable limitations. Our forecasting framework does not include several key state variables required by the full equations—such as density and pressure fields. As these quantities are not provided as model inputs, enforcing the full set of equations would be ill-posed and may introduce inconsistency rather than improving realism.
>
> For these reasons, we adopt the reduced-form momentum equations appropriate for the available data and the scope of the task.

---

### Author Response · Authors · 2025-11-30
**Summary our manuscript and rebuttal**

Dear PC, SAC, AC,

Thank you for reviewing this manuscript. As the deadline for the authors' response approaches, we have summarized this manuscript and rebuttal as follows.

**Summary of the manuscript:** M$^2$F-PINN adaptively learns multi-scale frequency-domain representations while jointly enforcing physical laws and data-based constraints. The overall framework consists of three stages: multi-frequency learning, PINN-based physical regularization, and data supervision. M$^2$F-PINN extends the applicability of PINNs to physical oceanography and offers a unique perspective on mitigating spectral bias in PINN training, thereby enhancing both the development and application potential of PINN-based models.

**Summary of the rebuttal:** We are pleased to have a positive discussion with everyone regarding the improvement of this manuscript. The overall rating of this manuscript is 2, 4, 2, 6. Although none of the four reviewers responded to our rebuttal, we would still like to express our sincere gratitude for his/her valuable feedback on our manuscript.

We are pleased to receive the following **positive feedback** from the reviewers: The paper systematically combines frequency-domain embeddings, physics-informed losses, and Transformer architectures in a coherent way.  - The NTK-based spectral analysis provides a clear explanation of how Fourier embeddings reshape learning rates across frequency bands, offering some theoretical insight beyond empirical results. **(Reviewers JjqD)** - M$^2$F-PINN achieves significantly lower Physical Inconsistency Coefficient (PIC) values, confirming that the predictions adhere more closely to the physical laws compared to all data-driven methods. - M$^2$F-PINN demonstrates state-of-the-art performance, achieving the lowest RMSE and highest ACC across short- and long-range forecasts (up to 60 days), outperforming both deep learning baselines and competitive ocean models (e.g., XiHe and WenHai). **(Reviewer 3uEE)** - The paper provides substantial theoretical support, explaining the impact of spectral bias on network training. - The ablation study examines the contribution of each major component (training B, training scale, training frequencies), which improves the transparency of the design. **(Reviewer 8DXa)** - Multi-scale representation via frequency embeddings. The use of Gaussian Fourier features helps alleviate spectral bias and improves learning of high-frequency dynamics, a common limitation of PINNs and spatiotemporal neural operators. - The results suggest that the multi-scale frequency-domain learning strategy, combined with physics-informed constraints, effectively enhances both prediction accuracy. **(Reviewer ZXEn)**

In addition, we actively respond to the comments of the reviewers. We have corrected the grammar, minor typo and other errors in the manuscript (Proposed by Reviewers JjqD and ZXEn). We conduct new experiments to validate the performance and power-spectrum analysis of M$^2$F-PINN (Proposed by Reviewers JjqD, 3uEE and 8DXa). We have added details to the article description to enhance its readability (Proposed by Reviewers JjqD, 3uEE, 8DXa and ZXEn). **We believe that through revisions to the manuscript and thorough communication with the reviewers, we have effectively addressed their concerns. At the same time, the overall quality of the manuscript has also been significantly improved.**

The above is our summary of this article and rebuttal. Thank you again to PC, SAC, and AC for their time and effort in
reviewing this manuscript.

Thanks and Regrads,

---

> ### Author Response · Authors · 2025-11-30
>
> ### Dear reviewers JjqD, 3uEE, 8DXa and ZXEn,
>
> Thank you to all the reviewers for your time and effort in reviewing this manuscript. Our author team carefully discussed the suggestions put forward and made revisions accordingly. These valuable suggestions greatly improved the quality of the manuscript, We believe this is a productive academic exchange that has benefited us greatly. Thank you again to the reviewers for their valuable feedback on this manuscript.
>
> Thanks and Regrads,

---

### Meta-Review · Area_Chair_cnAU · 2026-01-07

**Summary:**

The paper introduces a fourier feature embeddings to capture multi-scale phenomena as well as a PINN module that regularizes with the momentum equations for velocities. Overall, the paper presents strong results compared to several baselines.

Reviewers raised a few concerns that I believe are still outstanding. After going through the paper, the reviews, and the rebuttal, these are the major concerns:
1. Weak baselines: Some baselines (CNN, RNN) are simple and while it is okay to have them they don't represent a sufficient baseline suite. Other baselines show marginal performance drops compared to the proposed model and it is unclear if increased training would close the gap (the training cost for FNOs and MeshGraphNet is ~1/10th)
2. Unclear physics consistency metrics: The authors acknowledge that the physics may be inexact and it serves more as a physical regularizer. It is unclear how much attention one should pay to the PIC errors if that is the case. Further, it is unclear how these are calculated for data-driven models (no PINNs) like the FNO given the large time steps involved (>= 1 day), rather than near instantaneous time derivatives. It is also unclear how to interpret values O(1000) for this metric while models still have low RMSE.
3. The reviewers added power spectra to quantify the multi-scale emulation of their model. However, I was unable to find the performance of the baselines on power spectra on the plots. Further, baselines like FNO allow control on frequency modes and it is unclear if such hyperparameter tuning is sufficient to bring them to the same performance.
4. Newer neural operator approaches are not considered such as training with diffusion or generative modeling-based strategies that are known to capture high-resolutions very well (fine-scale).
5. The paper is not polished in its presentation (unclear fig/table captions, inconsistent significant figures, minimal explanation of what the figures are telling the reader).

Overall, the paper proposed a well-constructed model with fourier embeddings as well as physics losses leading to a novel contribution. However, the results are not comprehensive and the comparisons to baselines is weak. The paper has more room for improvement.

**Reviewer Concerns:**

Apart from the outstanding concerns highlighted above, the authors addressed other questions on the validity of the equations used, more baselines, added training / inference costs, added more diagnostics and RMSE values requested by the reviewers.

**Reviewer Scores:**

The scores stand at 2,4,2,6. I believe that reviewers may have come to a consensus at 4/5, but the outstanding concerns may have kept them at borderline.

---

### Decision · Program_Chairs · 2026-01-26

Reject